# Gut colonisation with multidrug-resistant *Klebsiella pneumoniae* worsens *Pseudomonas aeruginosa* lung infection

Rémi Le Guern [1] ✉, Teddy Grandjean [1], Sarah Stabler[1], Marvin Bauduin[1], Philippe Gosset [1], Éric Kipnis[1] & Rodrigue Dessein[1]

Carbapenemase-producing *Enterobacterales* (CPE) are spreading rapidly in hospital settings. Asymptomatic CPE gut colonisation may be associated with dysbiosis and gut-lung axis alterations, which could impact lung infection outcomes. In this study, in male C57BL/6JRj mice colonised by CPE, we characterise the resulting gut dysbiosis, and analyse the lung immune responses and outcomes of subsequent *Pseudomonas aeruginosa* lung infection. Asymptomatic gut colonisation by CPE leads to a specific gut dysbiosis and increases the severity of *P. aeruginosa* lung infection through lower numbers of alveolar macrophages and conventional dendritic cells. CPE-associated dysbiosis is characterised by a near disappearance of the *Muribaculaceae* family and lower levels of short-chain fatty acids. Faecal microbiota transplantation restores immune responses and outcomes of lung infection outcomes, demonstrating the involvement of CPE colonisation-induced gut dysbiosis in altering the immune gut-lung axis, possibly mediated by microbial metabolites such as short-chain fatty acids.

*Pseudomonas aeruginosa* is responsible for severe and deadly pneumonia in hospitalised patients, particularly in the intensive care unit. Outcomes of pneumonia caused by *P. aeruginosa* are not only determined by susceptibility/resistance to antimicrobials but also by pathogen virulence and the host immune response[1]. Recently, the gut microbiome and its alteration (dysbiosis) have emerged as components that alter the immune responses to lung infection, a paradigm referred to as the gut-lung axis[2].

Several studies in murine models of acute bacterial lung infection suggest that outcomes are determined by lung immune responses, which may be altered by gut dysbiosis[3]. Indeed, gut dysbiosis induced by broad-spectrum antibiotics worsens *P. aeruginosa* outcomes through altered lung IgA response[4]. Similarly, focusing on *Streptococcus pneumoniae* infection, antibiotic-induced dysbiosis resulted in impaired lung macrophage function and poor outcomes[5]. We also showed that antibiotic-induced gut dysbiosis worsened *P. aeruginosa* lung infection outcomes through a widely depressed lung cellular immune response[6]. These models rely on severely depleted or altered gut microbiota using broad-spectrum antibiotics.

Besides antibiotics, gut colonisation by multidrug-resistant bacteria itself has been associated with gut dysbiosis[7]. Of note, gut colonisation by carbapenemase-producing *Enterobacterales* (CPE) was associated with a 1.79 higher risk of mortality and an increased length of stay in patients admitted to the intensive care unit[8]. Furthermore, an increased abundance of *Enterobacterales* in the gut was associated with higher all-cause mortality in the general population, regardless of antimicrobial resistance[9]. In this study, death from respiratory causes was also increased[9]. Thus, beyond being a source of subsequent antibiotic-resistant infection, CPE colonisation itself might impact lung infection outcomes.

In this work, we show that in a murine model, CPE gut colonisation alters the lung immune response to *P. aeruginosa* infection and worsens outcomes by modulating the gut-lung axis differently from antibiotic-induced dysbiosis.

[1]Univ. Lille, CNRS, Inserm, CHU Lille, Institut Pasteur de Lille, U1019 - UMR 9017 - CIIL - Center for Infection and Immunity of Lille, F-59000 Lille, France.
✉e-mail: remi.leguern@chu-lille.fr

## Results

### Alterations of gut microbiota associated with antibiotic and CPE exposure

As previously reported[10], antibiotic administration (clindamycin) was required to maintain stable gut colonisation by CPE up to day 14 (Fig. 1b). In mice only exposed to CPE without antibiotic administration, CPE load was transient and fell below the limit of detection in stools as soon as five days after exposure. Mice exposed to both CPE and clindamycin presented neither symptoms of infection nor histological signs of colitis and were, therefore, colonised rather than infected (Fig. 1c).

We characterised the gut microbiota alterations associated with antibiotic (clindamycin) or CPE exposure. The gut microbiome of control mice and mice exposed to CPE only (without antibiotics) were similar 14 days after CPE exposure in terms of composition and α-diversity (Fig. 1d, e). Mice colonised by CPE (clindamycin and CPE exposure) presented the lowest α-diversity index and important modifications in microbiota composition. *Muribaculaceae* (formerly termed S24-7), a family of bacteria within the order *Bacteroidales*, represented 35.5% of the abundance of control mice, 25.9% after CPE exposure alone, 17.4% following administration of clindamycin and 0.03% in case of CPE colonisation (clindamycin and CPE exposure). *Muribaculaceae* were significantly decreased in CPE colonised mice compared to mice exposed to clindamycin alone ($p < 0.0001$) (differential abundance testing by DESeq2 in Supplementary Tables 1, 2). Conversely, the relative abundance of *Akkermansiaceae* increased in mice colonised by CPE compared to controls ($p < 0.0001$), but this was not statistically significant compared to mice exposed to clindamycin ($p = 0.56$). Gut colonisation by CPE was related to a specific dysbiosis characterised by a consistently marked decrease in *Muribaculaceae*, *Rikenellaceae*, and Lachnospiraceae_NK4A136_group compared to gut dysbiosis associated with clindamycin alone. Relative abundance in *Enterobacteriaceae* was increased in mice exposed to clindamycin or colonised by CPE compared with controls (Supplementary Table 1). At the genus level, *Escherichia/Shigella* were increased in mice exposed to clindamycin compared to controls, whereas both *Klebsiella* and *Escherichia/Shigella* were increased in CPE colonised mice compared to controls (Supplementary Table 2). By conventional microbiological culture, *Escherichia coli* levels were increased in mice exposed to clindamycin (8.7 log colony-forming units (CFU)/g of stool) or colonised by CPE (8.2 log CFU/g of stool) compared with controls (3.7 log CFU/g of stool), whereas *K. pneumoniae* was only recovered in mice colonised by CPE (8.9 log CFU/g of stool).

### Prior gut colonisation by CPE is associated with worse outcomes of subsequent *P. aeruginosa* lung infection

To assess the impact of antibiotic or CPE colonisation on the host defence against lung infection, mice were infected by intranasal instillation of *P. aeruginosa* on day 14 and evaluated 18 h later. Outcomes of lung infection by *P. aeruginosa* were worsened in mice colonised by CPE as assessed by clinical disease severity score, alveolar-capillary permeability (lung injury index) (Fig. 2a) and lung histological score (Fig. 2d), whereas the cell number in the bronchoalveolar lavage (BAL) did not differ (Fig. 2c). Histologic analysis revealed that colonisation by CPE promoted an inflammatory infiltrate in the peribronchial and alveolar areas (Fig. 2e). Moreover, it strongly increased bleeding area, alveolar wall thickening and lesions in the bronchial epithelium compared to infected controls. Treatment with clindamycin also increased the histologic score mainly by amplifying inflammatory cell recruitment and alveolar wall thickening, although with a lower intensity than in mice colonised by CPE. Bacterial burden did not significantly differ in the lung regardless of group (Fig. 2b). However, mice colonised by CPE presented increased dissemination of *P. aeruginosa* to the spleen. There was no significant difference in lung infection severity for mice exposed to CPE only (without antibiotics) or antibiotics only (without CPE).

To ensure that the worse lung infections outcomes were not specific to the CPE strain used (*K. pneumoniae* producing NDM carbapenemase), we reproduced these results with two other strains (*K. pneumoniae* producing OXA-48 or KPC carbapenemases) (Supplementary Fig. 1). Stable gut colonisation by CPE was maintained up to day 14. Clinical disease severity score, alveolar-capillary permeability (lung injury index), and dissemination of *P. aeruginosa* into the spleen were increased for mice colonised by *K. pneumoniae* producing OXA-48 or KPC carbapenemases. The worse outcomes of *P. aeruginosa* lung infection in mice colonised by CPE without any increase in bacterial load suggest that the host response could be responsible for the increased dissemination of the bacteria.

### Decrease of alveolar macrophages (AM) and dendritic cells (DC) in *P. aeruginosa*-infected lungs following gut colonisation by CPE

We evaluated the respective effects of prior antibiotic (clindamycin), CPE exposure alone or CPE colonisation (clindamycin and CPE exposure) on lung and spleen immune cellular responses to subsequent lung infection by *P. aeruginosa*. Infection by *P. aeruginosa* significantly increased the number of type 1 and 2 conventional dendritic cells (cDC1, cDC2) and natural killer T cells (NKT) compared to non-infected mice (intranasally instilled with sterile phosphate-buffered saline (PBS)). Cell population analysis on whole lung tissue revealed decreased AM and type 2 conventional dendritic cells (cDC2) only following prior gut colonisation by CPE (Fig. 3a). The class II MHC molecule expression in AM was also decreased in CPE mice compared to control infected mice but not in DC. Neutrophil numbers did not significantly differ. In the spleen, prior CPE colonisation decreased the number of macrophages, whereas other major populations of antigen-presenting cells and lymphocytes were not significantly modified (Supplementary Fig. 2). The concentrations of TNF-α in the BAL were significantly increased in mice colonised by CPE compared to mice having received clindamycin ($958.2 \pm 265.5$ versus $514.2 \pm 512.4$ pg/ml, $p = 0.048$) (Fig. 3b).

### Faecal microbiota transplantation improves *P. aeruginosa* lung infection outcomes following prior gut colonisation by CPE

Faecal microbiota transplantation (FMT) in CPE colonised mice limited the consequences of *P. aeruginosa* lung infection and restored outcomes similar to those of infected control mice (Fig. 4a–c). Namely, clinical disease score, alveolar-capillary permeability, BAL cell numbers and mortality were significantly lower when comparing colonised mice treated with FMT to colonised mice without FMT. In addition, FMT also decreased the histologic score in comparison with colonised mice (Supplementary Fig. 3). Indeed, FMT partly reduced the impact of CPE colonisation on bleeding areas, alveolar wall thickening and tissue lesions, whereas the inflammatory infiltrate is similar in both groups. Survival analysis showed lower mortality after FMT compared with mice colonised by CPE (60.0% versus 93.3%, $p = 0.03$) (Fig. 4d). These results demonstrate that the dysbiosis induced by CPE colonisation is involved in the observed phenotype of worsened *P. aeruginosa* lung infection.

We determined whether FMT could restore the host response to *P. aeruginosa* infection by assessing the immune response. Numbers of AM, cDC1 and cDC2 were partially restored following FMT, whereas their activation levels remained unchanged (Fig. 5a). Neutrophils did not significantly differ. In the spleen, FMT did not significantly modify the antigen-presenting cells, whereas it induced a degree of cell activation, as revealed by increased Class II MHC expression (Supplementary Fig. 4). In addition, FMT significantly decreased the concentrations of TNF-α in the BAL compared to colonised mice ($335.2 \pm 257.6$ versus $720.0 \pm 172.1$ pg/ml, $p < 0.001$) and IL-22 in comparison to control mice ($31.6 \pm 29.3$ versus $52.8 \pm 32.3$ pg/ml, $p = 0.04$) (Fig. 5b).

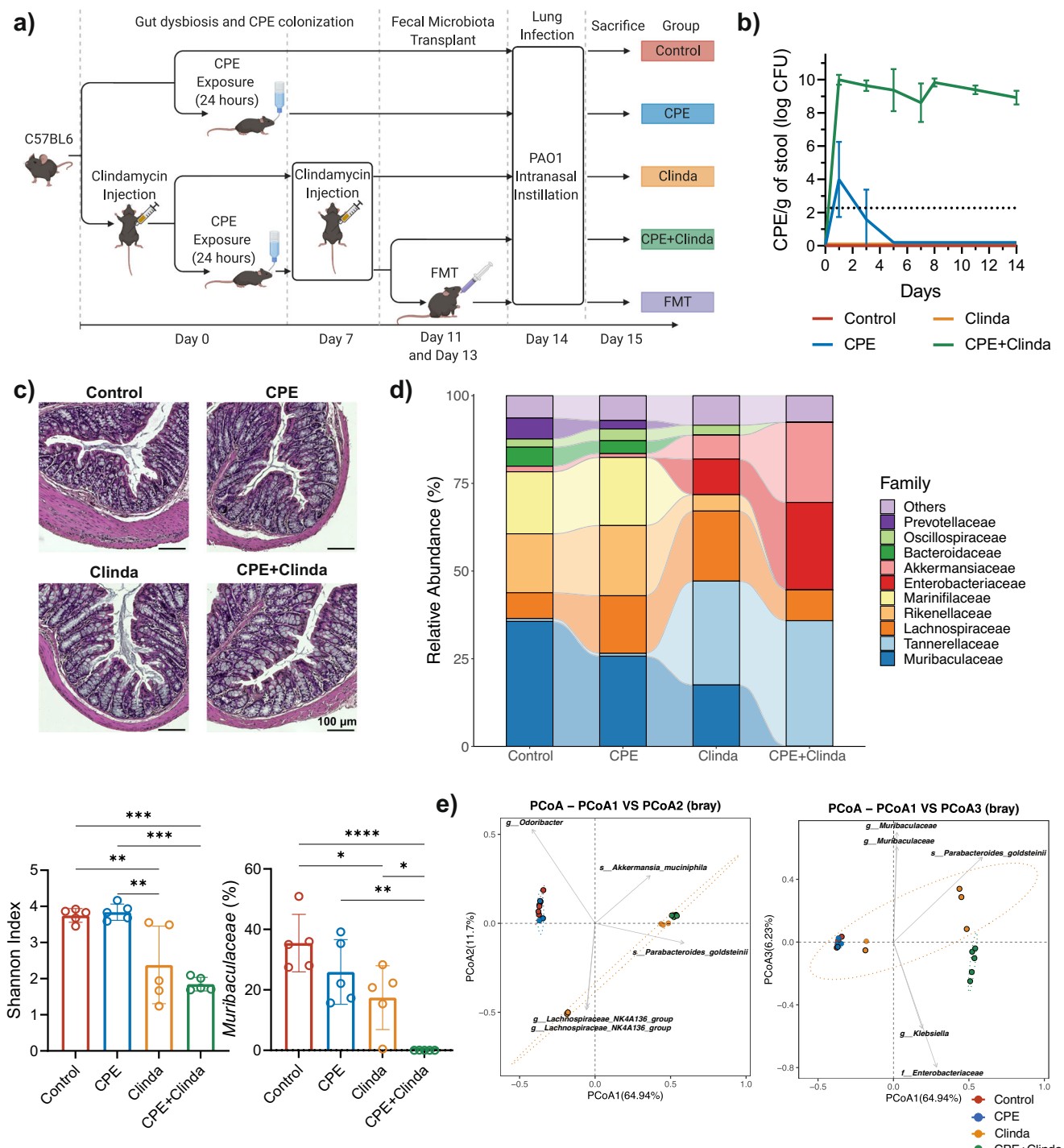

**Fig. 1 | Asymptomatic gut colonisation by carbapenemase-producing *Enterobacterales* (CPE) leads to a specific gut dysbiosis compared to antibiotics alone. a** Experimental plan and groups overview (created with BioRender.com). **b** Quantification of CPE load in stools for 14 days ($n = 13$ mice/group). **c** Representative images of colon sections stained by haematoxylin and eosin on day 14 (representative images out of 5 mice/group). Scale bars: 100 μm. **d** Gut microbiota 16S rRNA analysis at day 14 ($n = 5$ mice/group from the same experiment): relative abundance (family level), α-diversity (Shannon index), relative abundance in *Muribaculaceae*. Data are presented as mean values ± SD of biologically independent samples from different mice. *$p < 0.05$; **$p < 0.01$; ***$p < 0.001$; ****$p < 0.0001$ (one-way ANOVA followed by Tukey's post-hoc tests, see the Source Data file for the exact *P*-values). **e** Principal coordinates analysis (PCoA) biplot of β-diversity (Bray-Curtis dissimilarity) ($n = 5$ mice/group).

## Faecal microbiota transplantation partially restores gut microbiota composition without reducing CPE colonisation levels in stools

We assessed whether FMT 'decolonised' the gut of CPE by evaluating CPE loads in stools. FMT did not reduce stool concentration in CPE (Fig. 6a). Mean CPE load at day 14 (three days after the first FMT) was

8.0 ± 0.7 log CFU/g of stool for mice colonised by CPE and 8.1 ± 0.6 log CFU/g of stool following FMT ($p = 0.54$). However, FMT modified the gut microbiota composition and increased α-diversity, as shown by 16 S rRNA amplicon analysis (Fig. 6b). FMT was associated with a partial restoration in the relative abundance of *Muribaculaceae*, representing 3.7% for the mice colonised CPE and 20.9% following FMT

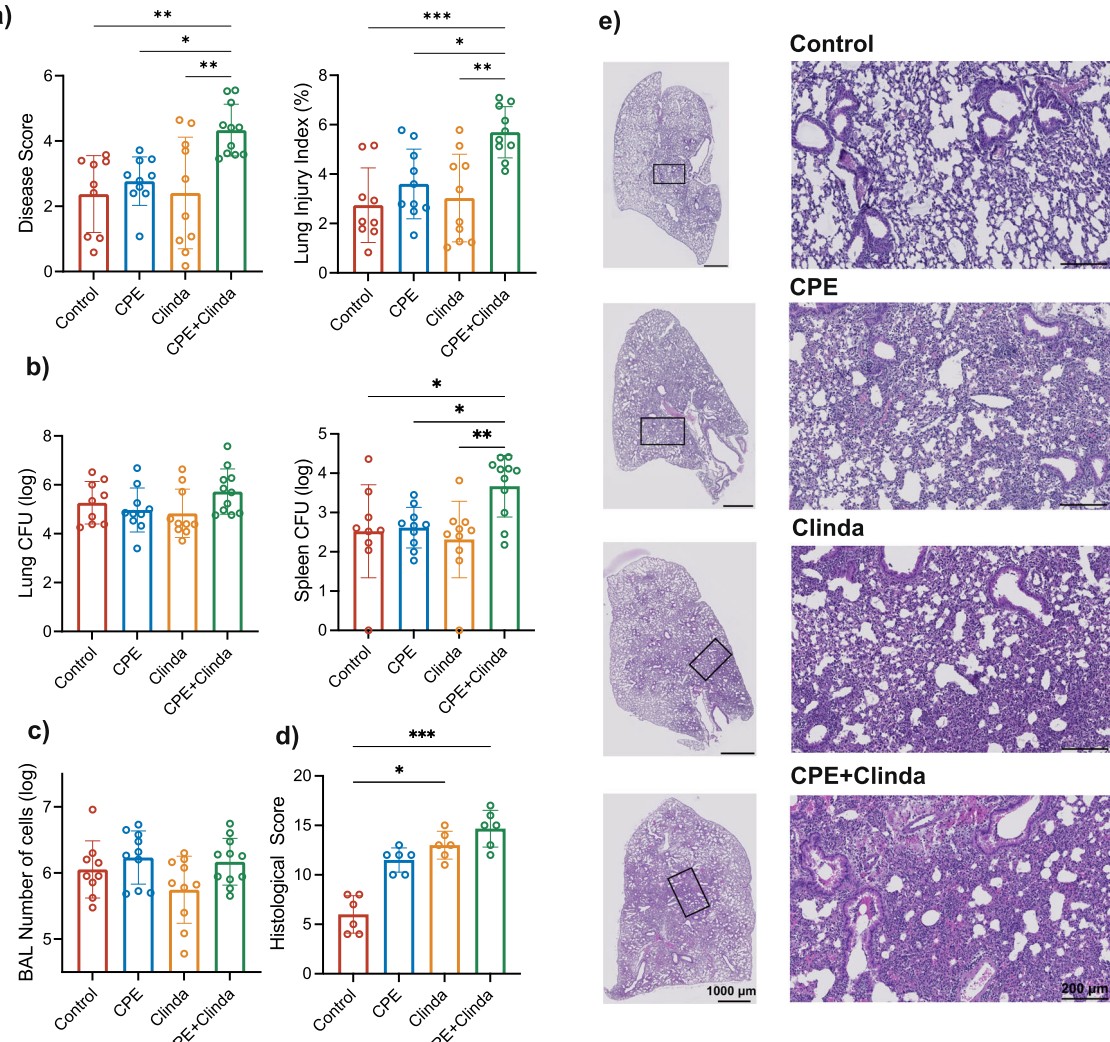

**Fig. 2 | Subsequent lung infection by *Pseudomonas aeruginosa* is more severe following gut colonisation by carbapenemase-producing *Enterobacterales* (CPE). a** Clinical disease severity score and alveolar-capillary permeability index ($n = 11$ mice/group). **b** Bacterial load in *P. aeruginosa* in the left lung and in the spleen ($n = 11$ mice/group). **c** Number of cells in the bronchoalveolar lavage fluid ($n = 11$ mice/group). **d** Histological score of lung infection ($n = 6$ mice/group). Data are presented as mean values ± SD of biologically independent samples from different mice. *$p < 0.05$; **$p < 0.01$; ***$p < 0.001$ (one-way ANOVA followed by Tukey's post-hoc tests, see the Source Data file for the exact *P*-values). **e** Representative images of lung tissue sections stained by haematoxylin and eosin (representative images out of 6 mice/group from two different experiments). Scale bars: 1000 µm (low magnification), 200 µm (high magnification).

($p < 0.001$) (differential abundance testing by DESeq2 in Supplementary Tables 3, 4). Conversely, the relative abundance of *Akkermansiaceae* tended to decrease following FMT compared with mice colonised by CPE (12.8% for CPE versus 4.8% for FMT), but this was not statistically significant ($p = 0.44$) (Supplementary Tables 3, 4). β-diversity (Bray-Curtis dissimilarity) showed partial restoration of the microbiota following FMT (Fig. 6c). Overall, FMT partially restored the gut microbiota diversity and composition in mice colonised by CPE without reducing CPE loads (i.e., without 'decolonising' the gut) in this time window.

### Supplementation in *Muribaculum intestinale* alone does not lead to stable colonization

Given the reduced abundance in *Muribaculaceae* for mice colonised by CPE, we evaluated the effects of supplementation in *Muribaculum intestinale* DSM 28989 by daily orogastric gavage for the three days preceding infection. Supplementation with *M. intestinale* did not significantly modify the worse lung infection outcomes associated with CPE colonisation (Supplementary Fig. 5). *M. intestinale* supplementation did not result in stable gut colonisation by *Muribaculum*, as the

relative abundance in *Muribaculum* sp. assessed by qPCR on Day 14 represented 7.9% for the control group, 0.009% for mice colonized by CPE, and 0.006% for mice supplemented in *M. intestinale* (Supplementary Fig. 5), in contrast with the data reported after FMT (Fig. 6).

### Gut short-chain fatty acids (SCFA) concentrations are reduced in CPE gut colonisation, and oral SCFA supplementation improves lung infection outcomes

Because *Muribaculaceae* abundance was increased following FMT and was previously linked with SCFA production[11], we investigated SCFA concentration in the caecum. Acetate, propionate, butyrate and total SCFA concentrations in caecum were decreased in mice colonised by CPE compared to those only exposed to clindamycin (total SCFA concentration $20.6 \pm 3.3$ µmol/g versus $39.0 \pm 16.3$ µmol/g, respectively, $p = 0.049$) (Fig. 7a). There was no significant difference in the total SCFA concentration after FMT, although there was a trend for increased levels (total SCFA concentration $20.6 \pm 3.3$ µmol/g for mice colonised by CPE versus $38.3 \pm 12.0$ µmol/g following FMT, $p = 0.063$). Nevertheless, FMT significantly increased the levels of acetate and propionate (Fig. 7a).

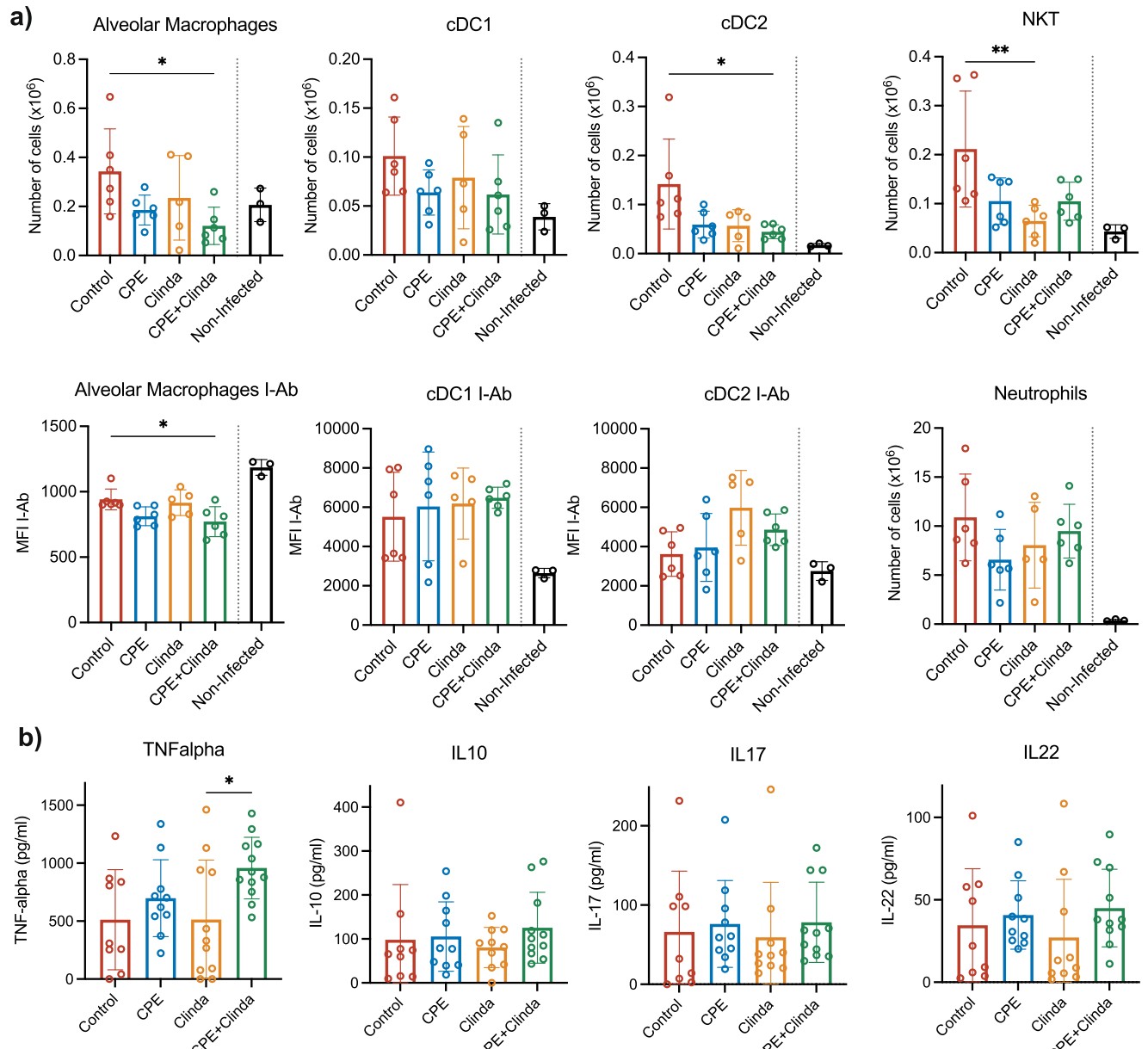

**Fig. 3 | Subsequent lung infection by *P. aeruginosa* is associated with decreased recruitment of alveolar macrophages (AM) and conventional dendritic cells (cDC) following gut colonisation by carbapenemase-producing *Enterobacterales* (CPE). a** Total numbers and activation of recruited cells in the lungs: antigen-presenting cells (alveolar macrophages and conventional dendritic cells), Natural Killer T cells, neutrophils (*n* = 6 mice/group for infected mice, *n* = 3 for non-infected mice). Non-infected mice correspond to control mice with nasal instillation of 50 μL of PBS instead of *P. aeruginosa*. MFI: median fluorescence intensity. **b** Cytokines in the bronchoalveolar lavage fluid (ELISA) (*n* = 12 mice/group). Data are presented as mean values ± SD of biologically independent samples from different mice. **p* < 0.05; ***p* < 0.01 (one-way ANOVA followed by Tukey's post-hoc tests, see the Source Data file for the exact *P*-values).

We investigated a potential therapeutic role of SCFA for mice colonised by CPE by supplementing drinking water in SCFA (acetate, propionate, and butyrate) for seven days before infection (experimental plan in Supplementary Fig. 6). SCFA supplementation significantly increased the concentrations in the caecal lumen for total SCFA and acetate, but there was no significant difference in propionate and butyrate levels (Supplementary Fig. 7).

SCFA supplementation did not significantly reduce stool concentration in CPE at day 14 (8.9 ± 0.3 log CFU/g of stool versus 9.1 ± 0.6 log CFU/g of stool *p* = 0.27).

Clinical severity and alveolar-capillary permeability were decreased following SCFA supplementation in mice infected with *P. aeruginosa* (Fig. 7b), whereas the bacterial load and BAL cell number

were unaltered (Fig. 7c, d). Survival analysis showed lower mortality following SCFA supplementation (Fig. 7e) (*p* = 0.01).

We analysed the lung immune profile of SCFA-treated mice by flow cytometry. SCFA had a limited effect on the recruitment of immune cells in the lungs and the spleen (Supplementary Figs. 8, 9).

## Discussion

Asymptomatic gut colonisation by CPE (*K. pneumoniae* NDM-1) is associated with decreased recruitment of AM and cDC and increased severity of subsequent *P. aeruginosa* lung infection. Colonisation with this exogenous multidrug-resistant bacteria leads to a specific gut dysbiosis compared to antibiotics alone. The restoration of alveolar immune cell recruitment and survival by FMT demonstrates the involvement of CPE colonisation-induced dysbiosis in altering the

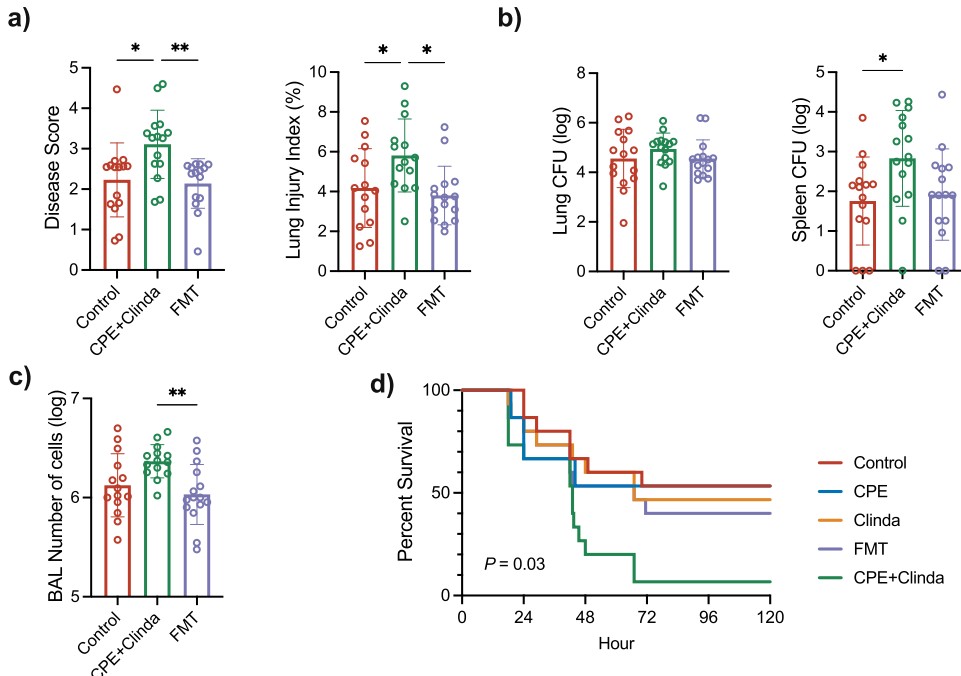

**Fig. 4 | Faecal microbiota transplantation (FMT) in mice colonised by carbapenemase-producing *Enterobacterales* (CPE) improves *P. aeruginosa* lung infection outcomes. a** Clinical disease severity score and alveolar-capillary permeability index ($n = 15$ mice/group). **b** Bacterial load in *P. aeruginosa* in the left lung and in the spleen ($n = 15$ mice/group). **c** Number of cells in the bronchoalveolar lavage fluid ($n = 15$ mice/group). Data are presented as mean values ± SD of biologically independent samples from different mice. \*$p < 0.05$; \*\*$p < 0.01$ (one-way ANOVA followed by Tukey's post-hoc tests, see the Source Data file for the exact *P*-values). **d** Log-rank (Mantel–Cox) analysis of survival curves ($n = 15$ mice/group).

immune gut-lung axis in response to *P. aeruginosa* lung infection. Alteration of the immune lung response by colonisation-induced gut dysbiosis involves microbial metabolites, including SCFA.

Alteration of the gut microbiota by antibiotics (clindamycin) was required to overcome colonisation resistance provided by the healthy gut microbiota and allow successful CPE colonisation, as previously described[12]. Indeed, mice exposed to CPE only (without antibiotics) were not colonised 14 days after CPE exposure and presented a gut microbiome similar to control mice. Alteration of the gut microbiota by clindamycin alone was associated with decreased α-diversity, expansion of *Tannerellaceae* and *Enterobacteriaceae* and decreased abundance in *Prevotellaceae*. Higher *Enterobacteriaceae* abundance in the clindamycin group compared to control was explained by an increase in the levels of commensal *Escherichia coli* as assessed by conventional microbiological culture. A hallmark of CPE colonisation compared to clindamycin alone was the near disappearance of *Muribaculaceae* and *Rikenellaceae*, associated with modifications in *Lachnospiraceae* (increase in *Hungatella* but decrease in Lachnospiraceae_NK4A136_group). *Muribaculaceae* and *Rikenellaceae* are families of bacteria within the order *Bacteroidales* belonging to the *Bacteroidetes* phyla. In another murine model, commensal *Bacteroidetes* indirectly protected against gut colonisation by carbapenemase-producing *K. pneumoniae* through IL-36 and macrophages[13].

*Akkermansiaceae* relative abundance increased in mice exposed to clindamycin (median 5.8%) or colonised by CPE (median 9.9%) compared with controls (median 0.1%) and tended to decrease after FMT (median 0.6%). *Akkermansia muciniphila* colonises the mucus of the gut and is considered to present beneficial anti-inflammatory effects[14]. However, an overabundance of *Akkermansia* (defined as > 4.8%) was associated with worse overall survival in patients with advanced cancer[15]. Moreover, antibiotic use was associated with a relative dominance of *Akkermansia* in this cohort[15]. In our model, an increase in *Akkermansia* may be explained by antibiotic use, but

further studies are warranted to evaluate the impact of *Akkermansia* overabundance on the worse lung outcomes.

Lung infection by *P. aeruginosa* was more severe in mice colonised by CPE in the gut, as assessed by an increased clinical score, alveolar-capillary permeability and mortality. AM and cDC were decreased in the lungs of mice colonised by CPE and infected by *P. aeruginosa*. AM are important actors of lung homeostasis[16], and phagocytosis is a critical process in the host defence against *P. aeruginosa* lung infection[17]. cDC are essential cells able to induce T cell activation and polarisation. This ability might explain the decreased mobilisation of NKT cells in colonised mice, as previously reported[18].

Several studies indicate a link between the gut microbiota and the host response to lung infection[4–6]. Specifically, the AM response to lung infection by *S. pneumoniae* was impaired in mice previously treated by broad-spectrum antibiotics for three weeks (targeting both the gut and lung microbiota)[5]. *P. aeruginosa* lung infection was also more severe following treatment by broad-spectrum antibiotics for 6 to 8 weeks[4]. In our study, antibiotic treatment alone (two injections of clindamycin) did not significantly increase *P. aeruginosa* lung infection severity seven days after the last injection of clindamycin. Possible explanations include the spectrum of the antibiotic, the shorter duration of the treatment used, and spontaneous resilience of the gut microbiota because subsequent lung infection occurred seven days after the last antibiotic administration in our model. Indeed, other published models use as many as four different antibiotics for a period of three weeks and wait only two days before lung infection[5]. Conversely, we showed that gut colonisation by CPE increased *P. aeruginosa* lung infection severity. One hypothesis could be that gut microbiota resilience is occurring in the clindamycin group but may be delayed in mice colonised by CPE.

We used faecal microbiota transplantation (FMT) to determine the involvement of CPE colonisation-induced dysbiosis in altered responses to *P. aeruginosa* lung infection. As expected, FMT partially restored the composition of the gut microbiome and increased the

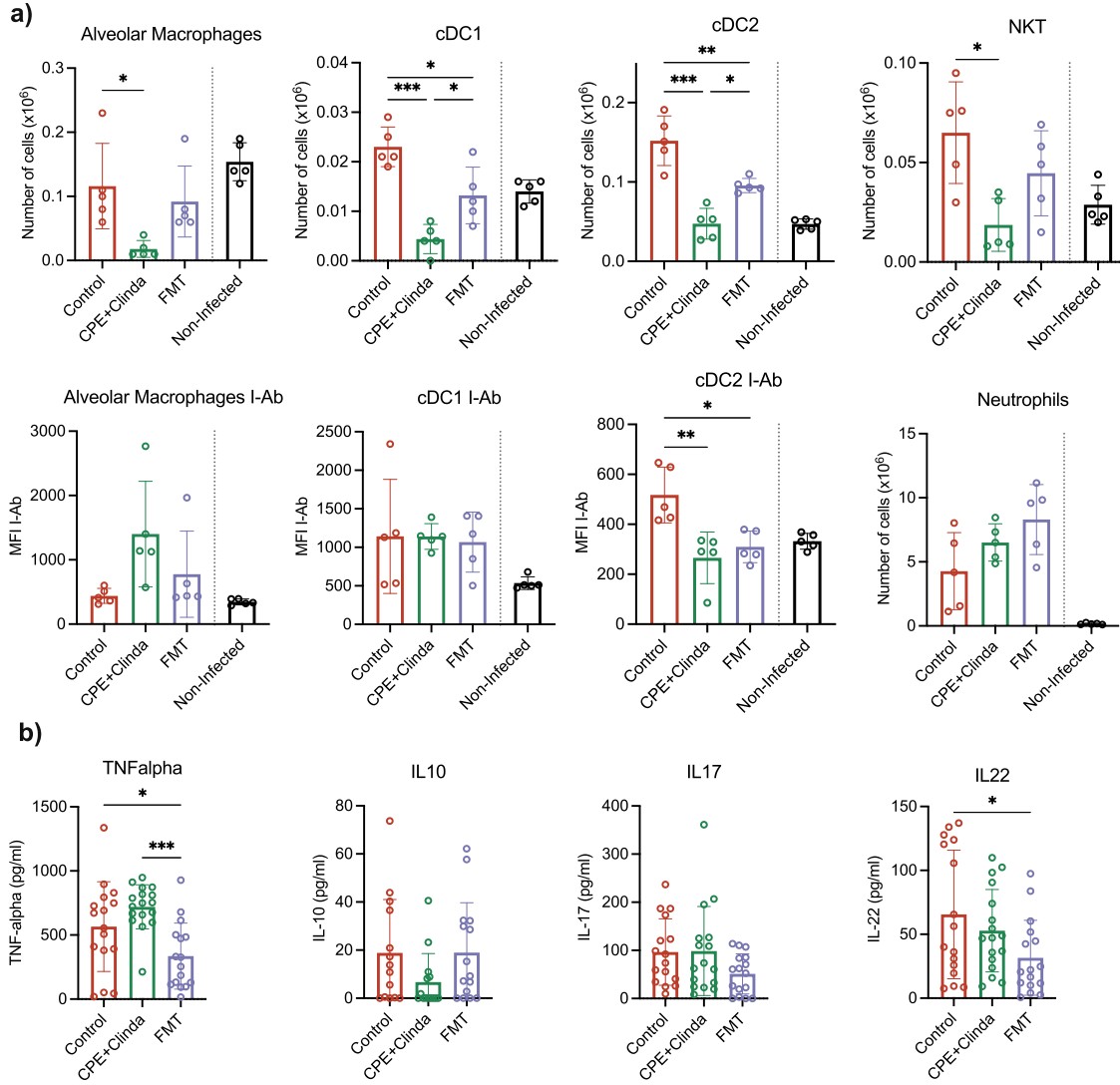

**Fig. 5 | Faecal microbiota transplantation (FMT) in mice colonised by carbapenemase-producing _Enterobacterales_ (CPE) partially restores lung antigen-presenting cells after lung infection by _P. aeruginosa_. a** Total numbers and activation of recruited cells in the lungs: antigen-presenting cells (alveolar macrophages and conventional dendritic cells), Natural Killer T cells, neutrophils (_n_ = 5 mice/group). Non-infected mice correspond to control mice with nasal instillation of 50 μL of PBS instead of _P. aeruginosa_. MFI: median fluorescence intensity. **b** Cytokines in the bronchoalveolar lavage fluid (ELISA) (_n_ = 17 mice/ group). Data are presented as mean values ± SD of biologically independent samples from different mice. \*_p_ < 0.05; \*\*_p_ < 0.01; \*\*\*_p_ < 0.001 (one-way ANOVA followed by Tukey's post-hoc tests, see the Source Data file for the exact _P_-values).

α-diversity of mice previously colonised by CPE. Notably, _Muribaculaceae_ re-expanded following FMT (20.9%) compared with mice colonised by CPE without FMT (3.7%). Of note, FMT did not act through 'decolonisation' of the gut because there was no CPE load decrease in stools within three days after the first administration. Therefore, the effects we observed in our model are mainly due to the specific dysbiosis associated with CPE implantation rather than the presence of CPE. Similarly, in patients colonised by multidrug-resistant organisms, FMT was recently associated with reduced bloodstream infections either due to multidrug-resistant organisms or for all microorganisms[19,20].

We tried to evaluate the effect of specific supplementation in _Muribaculum intestinale_ by iterative orogastric gavage but did not obtain stable gut colonisation with _M. intestinale_. Recently, Chng et al. evaluated the gut microbiota recovery following antibiotic therapy in a murine model[21]. They showed that most of the bacterial species producing SCFA were considered secondary of tertiary species that need breakdown products from primary species. In our model, colonisation was possible following FMT because a diverse ecosystem was provided, confirming that _Muribaculaceae_ are not able to colonise the gut when they are given individually.

The relationship between colonisation-induced gut dysbiosis and lung infection severity may involve microbial metabolites such as short-chain fatty acids (SCFA), which were reported to protect against lung infections[2,22]. In vitro, SCFA also directly inhibited the growth of antibiotic-resistant _Enterobacterales_, including carbapenemase-producing _K. pneumoniae_[23]. SCFA gut concentrations were the lowest in mice colonised by CPE, possibly related to reduced _Muribaculaceae_ abundance. Indeed, _Muribaculaceae_ were previously correlated with gut SCFA concentrations[11,24]. Nevertheless, several other bacterial families in the gut can also be involved in SCFA production: acetate is produced by multiple enteric bacteria (_Prevotella_, _Bacteroides_, _Akkermansia_), propionate by several _Bacteroidetes_ (_Bacteroidaceae_, _Prevotellaceae_) and butyrate by _Firmicutes_ (_Faecalibacterium_, _Eubacterium_)[25,26]. Interestingly, we found that SCFA supplementation for seven days was associated with decreased lung infection severity and mortality in mice colonised by CPE, showing a potential role as a therapeutic target. Schulthess et al. reported that

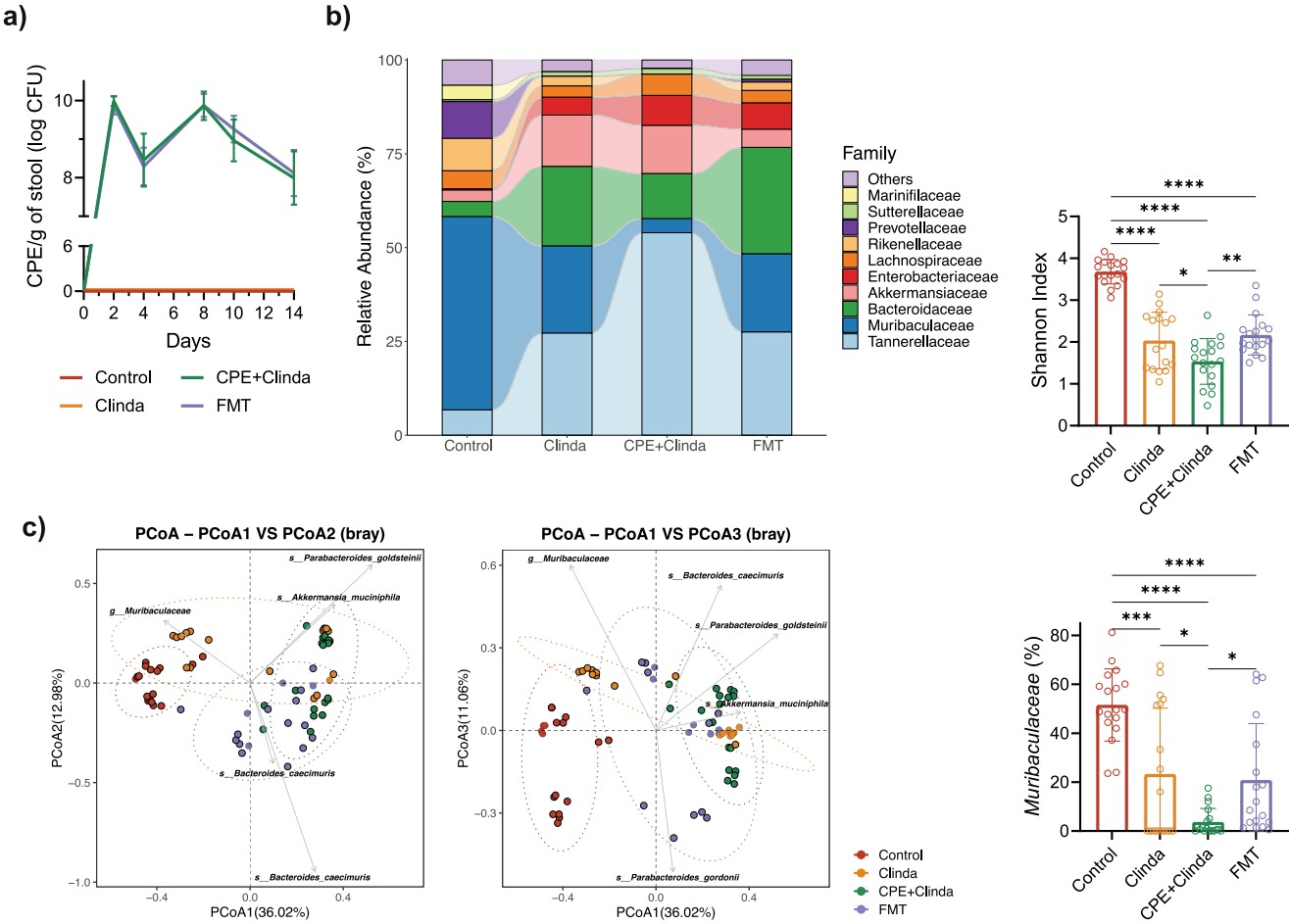

**Fig. 6 | Faecal microbiota transplantation (FMT) partially restores gut microbiota composition without reducing carbapenemase-producing _Enterobacterales_ (CPE) colonisation levels in stools. a** Quantification of CPE load in stools (n = 20 mice/group). **b** Gut microbiota 16 S rRNA analysis (n = 18 mice/group, representatives from 4 experiments): relative abundance (family level), α-diversity (Shannon index), relative abundance in _Muribaculaceae_. Data are presented as mean values ± SD of biologically independent samples from different mice. *$p < 0.05$; **$p < 0.01$; ***$p < 0.001$; ****$p < 0.0001$ (one-way ANOVA followed by Tukey's post-hoc tests, see the Source Data file for the exact P-values). **c** Principal coordinates analysis (PCoA) biplot of β-diversity (Bray–Curtis dissimilarity) (n = 18 mice/group).

macrophages differentiated in the presence of butyrate present increased antimicrobial activity against bacterial pathogens[27]. Moreover, propionate and butyrate increased the barrier of airway epithelial cells[28]. In our model, SCFA restored epithelial barrier function (as assessed by alveolar-capillary permeability) without modification of the bacterial load. SCFA had a limited effect on the recruitment of immune cells in the lungs and the spleen, in contrast with FMT. Our data suggest that SCFA supplementation restored an efficient antibacterial response, possibly by activating the resident cells, and limited the alteration of the alveolar-capillary barrier rather than by the recruitment of immune cells. SCFA supplementation is not the only option to increase SCFA levels. Specifically, dietary fibre, probiotics, or a defined microbial consortium can also increase gut SCFA production[29,30].

This study has several limitations. First, we focused only on _K. pneumoniae_ and did not evaluate the effects of colonisation with other members of the _Enterobacteriaceae_ family. However, we evaluated three strains of _K. pneumoniae_, including an NDM-producing strain belonging to ST-147, which has an epidemic potential to disseminate in healthcare settings[31]. Secondly, to assess the mechanism of increased lung infection severity, we specifically analysed SCFA concentrations rather than untargeted metabolomics, whereas other metabolites are potentially involved in the gut-lung axis (e.g., tryptophan metabolites)[32]. Despite these limitations, we showed a decrease in lung

infection severity following SCFA supplementation, highlighting SCFA's role in potential therapeutic modulation.

Overall, asymptomatic colonisation by CPE was associated with increased severity and mortality of subsequent _P. aeruginosa_ lung infection through specific alterations of the gut microbiota. We demonstrated the deleterious impact of asymptomatic gut CPE colonisation on outcomes of subsequent lung infection. The effects of FMT suggest that colonisation-induced gut dysbiosis (rather than the presence of CPE) drives the deleterious consequences on lung infection response. SCFA supplementation or other therapeutic interventions leading to SCFA production may be helpful in the case of persistent gut dysbiosis to limit subsequent lung infection severity.

## Methods

### Bacterial strains

A clinical strain of _Klebsiella pneumoniae_ producing NDM-1 carbapenemase (KPL0.2) was used for CPE gut colonisation for all the experiments except Supplementary Fig 1. KPL0.2 was whole-genome sequenced and deposited into Genbank[33]. Using Kleborate[34], the virulence score of KPL0.2 was low (1 out of 5, with only the presence of yersiniabactin). Two other strains of carbapenemase-producing _K. pneumoniae_ were used for Supplementary Fig. 1: KPL0.1 (OXA-48 carbapenemase)[33] and KPL0.3 (KPC carbapenemase) (Genbank accession number NZ_JANLCP000000000). _Pseudomonas aeruginosa_ strain

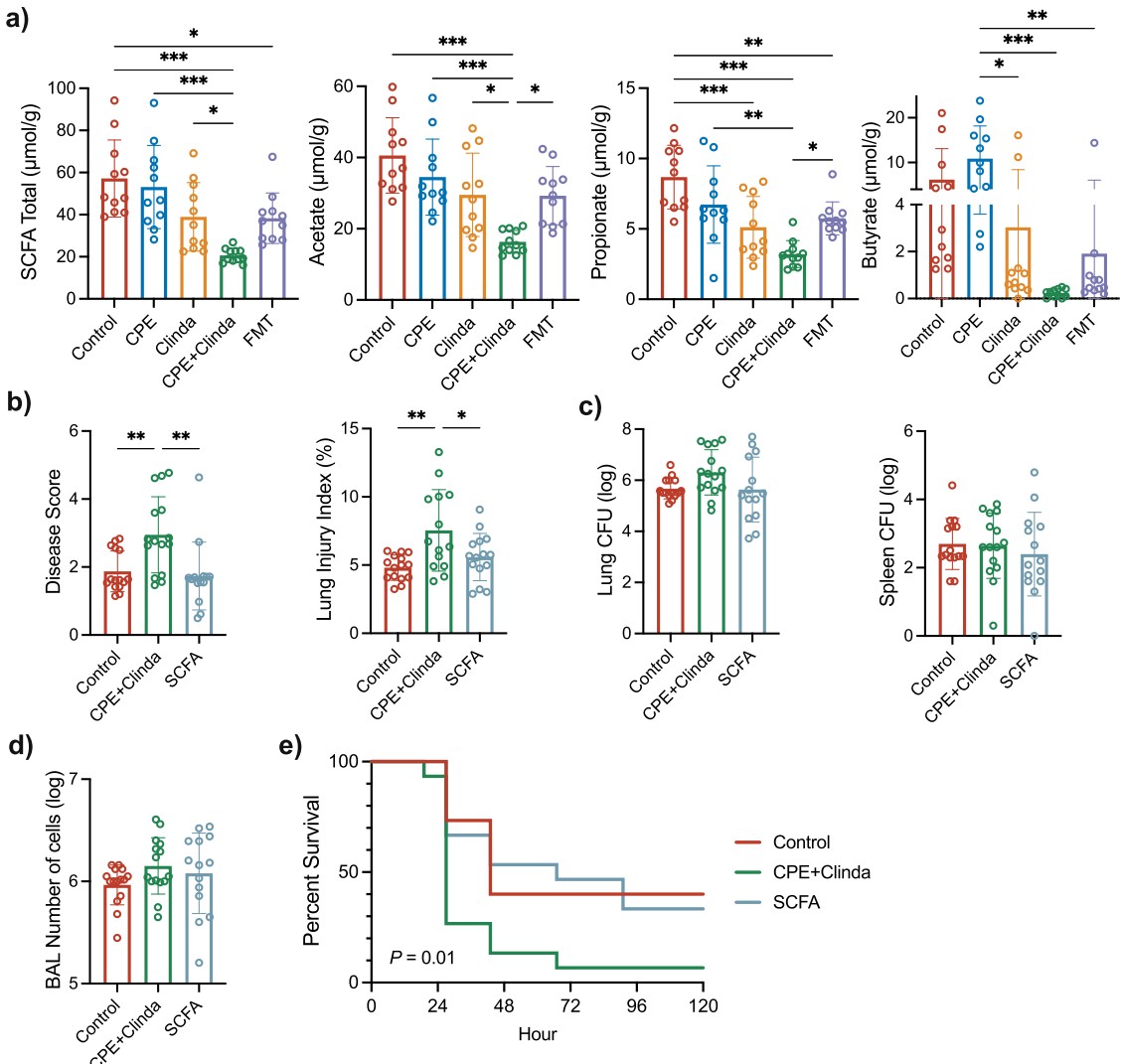

**Fig. 7 | Gut short-chain fatty acids (SCFA) concentration is reduced in case of carbapenemase-producing *Enterobacterales* (CPE) colonisation, and supplementation by a blend of SCFA improves lung infection outcomes.**
**a** Quantification of SCFA in caecum lumen samples ($n = 11$ mice/group) at day 14. Supplementation in SCFA (acetate 100 mM, propionate 25 mM, butyrate 25 mM in drinking water) for seven days in mice colonised by CPE improves subsequent *P. aeruginosa* lung infection outcomes: (**b**) clinical disease severity score, alveolar-capillary permeability index ($n = 15$ mice/group). **c** Bacterial load in *P. aeruginosa* in the left lung and in the spleen ($n = 15$ mice/group). **d** Number of cells in the bronchoalveolar lavage fluid ($n = 15$ mice/group). Data are presented as mean values ± SD of biologically independent samples from different mice. *$p < 0.05$; **$p < 0.01$; ***$p < 0.001$ (one-way ANOVA followed by Tukey's post-hoc tests, see the Source Data file for the exact $P$-values). **e** Log-rank (Mantel−Cox) analysis of survival curves ($n = 15$ mice per group).

PAO1 was used for lung infection. *K. pneumoniae* and *P. aeruginosa* are naturally resistant to clindamycin. Bacteria were stored at −80 °C in 30% glycerol (vol/vol), maintained on LB agar plates, and cultured overnight in Lysogeny Broth Lennox (LB) at 37 °C with orbital shaking at 100 rpm. To ensure growth in mid-log phase, a 1/40 dilution of the overnight culture was prepared and incubated for 2 h at 37 °C with shaking. Bacteria were washed twice with sterile phosphate-buffered saline (PBS). The bacterial pellet was resuspended in PBS, and the optical density was measured at 600 nm; the desired inoculum was obtained by dilution. Bacterial suspensions were verified by serial dilution and plating on LB agar plates.

*Muribaculum intestinale* DSM 28989 (type strain)[35] was bought from DSMZ. *M. intestinale* was plated onto VL agar supplemented with cysteine and incubated anaerobically for seven days.

## Mice

Male C57BL/6JRj mice aged 6 to 8 weeks were purchased from Janvier Labs and housed under specific pathogen-free conditions. Each group was kept in its own cage (maximum of five mice per cage), and all cages were changed twice per week. Mice had free access to a standard laboratory food diet and water, and were housed under a 12 h light-dark cycle, 50−70% humidity and 20−24 °C temperature. The animal procedure followed in this study was in accordance with the French Guidelines for the Care and Use of Laboratory Animals and the European Union guidelines. The current project has been approved by the National Institutional Animal Care and Use Committee (CEEA 75) and received the authorisation number APAFIS #7166.

## Experimental study design

The experimental design and main study groups are illustrated in Fig. 1a. Gut dysbiosis was induced by intraperitoneal clindamycin injection (10 mg/kg) on days 0 and 7. Mice were exposed to CPE in drinking water ($1 \times 10^7$ CFU/mL) for 24 h from days 0 to 1 and then switched to tap water. When indicated, faecal microbiota transplantation was performed on days 11 and 13. When indicated, orogastric gavage with *M. intestinale* was performed on days 11, 12 and 13. Stool samples were collected from control mice, diluted (4 faecal pellets in 1 mL of PBS), homogenised by bead-beating using a TissueLyser II

(Qiagen, Hilden, Germany) with ceramic beads of 1.4 mm, and filtered through a 70 μm cell strainer. Mice received 200 μL of the filtrate via oral gavage. When indicated, short-chain fatty acid (SCFA) supplementation in drinking water occurred from days 8 to 14, with a blend of acetate (100 mM), propionate (25 mM), and butyrate (25 mM) changed twice weekly. An SCFA concentration of 150 mM did not decrease water oral intake, as previously reported[36].

### Quantification of bacteria
Stool samples were collected directly from each mouse during bowel movements, weighed and homogenised by bead-beating using a TissueLyser II (Qiagen, Hilden, Germany) with ceramic beads of 1.4 mm. CPE load in stool samples was evaluated after serial dilution by plating 100 μl onto a selective medium (LB agar with 32 mg/litre cefotaxime and 6 mg/litre vancomycin).

To evaluate the lung and spleen bacterial loads of *Pseudomonas aeruginosa*, the left lobe of the lung and spleen were collected and homogenised by bead-beating, serially diluted, and then 100 μl were plated onto non-selective (LB agar without antibiotics) and selective medium for *P. aeruginosa* (Cetrimide agar).

### Model of infectious pneumonia
On day 14, acute lung infection was induced by intranasal instillation. Mice were anaesthetised by inhaling 3–5% isoflurane using a vaporiser; then, 50 μL of *P. aeruginosa* PAO1 ($1 \times 10^7$ CFU) or 50 μL PBS for non-infected mice were administered by intranasal instillation. For survival analysis, mice were intranasally infected with 50 μL of *P. aeruginosa* PAO1 ($2 \times 10^7$ CFU) and monitored for 120 h.

### Clinical score
At 18 h post-infection, mice were scored for severity of pneumonia on a clinical score developed in our laboratory, based on previously published parameters[37]. Mouse behaviour, appearance and surface temperature were evaluated by a blinded observer. Fur aspect: 0 = normal coat, 1 = slightly ruffled fur, 2 = ruffled fur. Activity: 0 = normal, 1 = reduced, 2 = little or none with provocation. Temperature: 0 = 30–35 °C, 2 = less than 30 °C. Clinical score = [(Fur aspect score + activity score + temperature score) + (percentage of weight loss/10)]

### Lung injury
Alveolar-capillary membrane permeability was evaluated using fluorescein isothiocyanate (FITC)-labelled albumin leakage from the vascular to the alveolar-interstitial compartment 18 h after infection. Intraperitoneal injection 200 μL of albumin-FITC at 2 mg/mL was performed 2 h before sacrifice, and blood and bronchoalveolar lavage (BAL) were then collected. For this, 1.5 mL of PBS was injected into the mice trachea, and the recovered BAL fluid was centrifuged at $200 \times g$ for 10 min. The supernatant was used to evaluate endothelial permeability and the pellet for cell counts. The fluorescence ratio measured in BAL, divided by serum fluorescence, reflected alveolar-capillary permeability.

### Flow cytometric analysis
Lung and spleen tissues were collected aseptically and analysed for CFU counts, histology, cytokine measurement and flow cytometry analysis. For this, the lungs were perfused with PBS, and the left lobe was treated with collagenase (Sigma-Aldrich). The leucocyte-enriched fraction was collected using a Percoll gradient (GE Healthcare) before flow cytometry staining and culture. Both lung and spleen cells were washed and incubated with antibodies (Supplementary Table 5) for 30 min in PBS before being washed. Staining was performed as previously described in online supplementary information reported by Sharan et al.[38]. Data were acquired on an LSR Fortessa (BD Biosciences) and analysed with FlowJo™ software v7.6.5 (Stanford, CA, USA). Debris was excluded according to size (FSC) and granularity (SSC). Immune

cells expressing CD45 were gated to analyse the frequency, activation and number of cell subsets. Phenotypes are shown in Supplementary Table 6, and the gating strategy is reported in Supplementary Fig. 10.

### Microbiome and short-chain fatty acids analysis
Total DNA was extracted from murine faeces using the QIAamp DNA Stool Mini Kit (QIAGEN, Courtabeuf, France). Negative controls were included to assess reagent and environmental contamination. The V6–V8 region of the 16 S rRNA gene was amplified using the primers B969F (ACGCGHNRAACCTTACC) and BA1406R (ACGGGCRGTGWGTRCAA)[39]. Sequencing was conducted on an Illumina MiSeq using v3 600 cycle chemistry at Dalhousie University's Integrated Microbiome Resource (IMR; imr.bio). Microbiome data were analysed using QIIME 2 2020.8. Raw sequence data were demultiplexed and quality filtered with the q2-demux plugin, followed by denoising with DADA2. Taxonomy was assigned to amplicon sequence variants (ASVs) using the q2-feature-classifier plugin against the SILVA 132 99% database. ASVs clustered in phylogenetic levels were further analysed using the phyloseq package in R software version 4.2.2 (R Core Team, Vienna, Austria) to perform α-diversity estimates and plot relative abundances. β-diversity was analysed with principal coordinate analysis (PCoA) of Bray-Curtis distances using the R package microbial. Differential abundance testing was performed using DESeq2 and applying Benjamini-Hochberg False Discovery Rate.

Relative abundance in *Muribaculum* sp. was evaluated by qPCR for Supplementary Fig. 7. Primers Mb-F (GAGAGTACCTGAAGAAAAAGC) and Mb-R (ACGCATTCCGCATACTTCT) were designed and used to amplify *Muribaculum* sp. Universal 16 S primers 16S-F (TCCTACGG-GAGGCAGCAGT) and 16S-R (GGACTACCAGGGTATCTAATCCTGTT) were used for total bacteria. PCR was performed in 20 μl reaction volume: 10 μL Brilliant III SYBR® Green QPCR mix (Agilent, CA, USA), 0.3 μl ROX, 5.7 μl DNAse free water, 1 μl of forward primer (2 μM), and 1 μl of reverse primer (2 μM). The amplification reaction was performed in AriaMX Real-Time PCR System (Agilent, CA, USA), with an initial denaturing step of 3 min at 95 °C and 40 cycles of 95 °C for 5″, 60 °C for 30″. *M. intestinale* strain DSM 28989 was included as a positive control. Relative abundance was calculated as previously reported[40].

Caecum lumen samples were dissolved in filter-sterilised water prior to SCFA determination. Acetate, propionate, butyrate, valerate, caproate and branched SCFAs (isobutyrate, isovalerate and isocaproate) were quantified as described previously[41].

### Histological analysis
Colon and lungs were fixed in formalin and then paraffin-embedded. Cross-sections were stained by haematoxylin-eosin. To quantify lung lesions, a quantitative histopathologic score was used[42].

### Statistical analysis
Statistical analysis was performed using GraphPad Prism 9.2 (GraphPad Software, La Jolla, CA, USA) and R 4.2.2 (R Core Team, Vienna, Austria). Continuous variables were compared using ANOVA followed by a Tukey post hoc test. All error bars correspond to standard deviations. Survival curves were analysed using the log-rank (Mantel−Cox) test. Animals were randomly assigned to groups before experimentation. *P*-values < 0.05 were considered statistically different.

### Reporting summary
Further information on research design is available in the Nature Portfolio Reporting Summary linked to this article.

## Data availability
Microbiome sequencing data have been deposited to Sequence Read Archive (BioProject PRJNA780243). Source data are provided with this paper.

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

## Acknowledgements

The authors thank André Comeau from the Integrated Microbiome Resource (imr.bio) for help with 16S rRNA analysis of the gut microbiota. Massimo Marzorati and Cindy Duysburgh (ProDigest) are acknowledged for help with SCFA quantification. We thank Antonino Bongiovanni and Hélène Bauderlique of the BioImaging Center Lille (BICeL) facility for access to systems and technical advice on microscopy and flow cytometry. Marie-Hélène Gevaert is acknowledged for technical help with histology. We thank Olivier Le Rouzic for critical reading of the manuscript.

## Author contributions

Conceptualisation: R.D., E.K., R.L.G.; Investigation: R.L.G., T.G., P.G., S.S., M.B.; Formal analysis: R.L.G.; Supervision: R.D., E.K., P.G.; Writing (original draft): R.L.G., P.G., E.K., R.D.; Writing (review & editing): all authors. Approval of the final manuscript: all authors.

## Competing interests

The authors declare no competing interests.
