## [Peer review file · Nature Communications]

REVIEWER COMMENTS

Reviewer #1 (Remarks to the Author):

In the manuscript titled “Gut colonisation with multidrug-resistant *Klebsiella pneumoniae* worsens *Pseudomonas aeruginosa* lung infection” authors seek to determine the effect of asymptomatic gut colonization with multidrug-resistant *Klebsiella pneumoniae* (CPE) on outcome of lung infection provoked by *P.aeruginosa*. The authors first established a model of stable gut colonization with CPE by treating mice with clindamycin and exposing them to CPE in drinking water. After this, they infect mice with *P. aeruginosa* and follow the infection progression. The authors characterized microbiome changes, disease and histological scores, and lung immune cells profile as a result of *P. aeruginosa* infection in CPE colonized mice. They saw that CPE colonized and *P. aeruginosa* infected mice had higher disease score than animals that weren’t CPE colonized. They also identified members of the gut microbiome whose abundances were altered in CPE colonized and *P. aeruginosa* infected mice which lead them to hypothesize that these bacteria (specifically Muribaculaceae) might be responsible for worse outcome of lung infection in CPE colonized animals. To test this, authors performed faecal microbiome transplant in CPE colonized and *P. aeruginosa* infected mice which resulted in lower disease score and mortality as compared to mice that were colonized and infected but didn’t receive FMT. FMT also affected lung immune profile. The authors further hypothesized that altered abundance of Muribaculaceae in the gut of CPE colonized mice was resulting in lower short-chain fatty acid (SCFA) abundance and was the reason for more severe lung infection in these animals. They tested the role of SCFA administration on mortality and disease score in CPE colonized and *P.aeruginosa* infected animals and saw reduction in both.

The role of gut microbiome dysbiosis on host’s health is a subject of great interest and different aspects of it have been widely studied. The authors take on this, by studying the effect of CPE gut colonization on outcome of *P.aeruginosa* provoked lung infection, is novel. However, there are several major points that would need to be addressed for the manuscript to be published.

Major points:

1. It would be important to know if the effect observed by CPE colonization of the gut on lung infection outcome is specific for the strain authors used or it could be reproduced with other members of Enterobacteriaceae family.
2. The authors were focusing on Muribaculaceae as potentially relevant microbiome members whose depletion led to worse lung infection outcomes. However, these bacteria are not the only ones with altered abundances in CPE colonized mice. The role of Akkermansiaceae and Sutterellaceae was not investigated nor discussed even though these were also altered in CPE colonized mice. In fact, Akkermansiaceae were increased in CPE colonized mice, but upon FMT their levels decreased, suggesting these can also explain the lung immune phenotype authors observed. Since members of this

family are known for colonizing the mucus of the gut and impacting the immune response of the host it is of high importance to investigate their role in this model.

3. The evidence pointing to SCFAs as responsible for lung phenotype and infection outcome is inconclusive: the administration of SCFAs to CPE colonized animals does have an ameliorating effect on infection outcome, however this seems to be unrelated to previously described results pertaining to this animal model. The animals that received clindamycin only (without CPE exposure) had similar levels of SCFA to CPE colonized animals although their outcome parameters were pronouncedly different, suggesting SCFA alone are not responsible for the higher mortality and disease score in CPE colonized mice. Moreover, it would be necessary to see lung immune profile in SCFA treated animals and compare it to CPE colonized and FMT treated animals to be able to make conclusions on whether SCFA are in fact restoring lung alterations caused by CPE colonization. Also, readers need to be provided with SCFA levels upon their administration so that those can be compared to SCFA levels in CPE colonized and FMT treated animals.

4. The relationship between Muribaculaceae abundance and SCFAs wasn't proved. The authors suggested that decrease in Muribaculaceae in CPE colonized animals was the reason for SCFA depletion. It would be relevant to perform experiments in which CPE colonized animals would be treated with a member of Muribaculaceae family and disease outcome and SCFA levels would be followed. In fact, this should be performed also for a member of Akkermansiaceae as well. Confirming the role of these gut bacteria on lung infection outcome before diving into the mechanism of action is very important for making solid conclusions.

Minor points:

1. Microbiome analysis could be more extensive. It would be nice if authors could provide comprehensive tables of all taxa passing FDR when different experimental groups are compared. This could be done for both figures 1 and 6 and provided as supplementary tables. Also, it would be easier for the reader to understand differences between microbiome of different experimental groups represented in Figure 1 if the authors provided PCoA biplot instead of 2 separate graphs with differences between CPE and CPE+clindamycin; and clindamycin and CPE+clindamycin. This biplot should include also control group of animals.

2. Could authors please provide p values in text when describing results of statistical analysis?

3. Authors should introduce all the figures in text. In many instances results are described without explaining where the result is plotted.

4. Authors should introduce properly all the abbreviations. In several instances abbreviation was never introduced, or it was introduced later in text.

5. In the results section authors mention alveolar capillary permeability, while in the figures they plot % of lung injury. Could the authors clarify in text that this is the same thing.

6. Results represented in Figure 3b were never mentioned nor explained in the text.

7. Authors suggested in text that CPE decontamination by FMT was unsuccessful, since CPE could be detected in feces a day after FMT. I suggest the authors to always refer to this time window, since it is unclear if the CPE levels would drop in FMT treated group in case the animals were followed beyond two days post FMT.
8. In Discussion, 2nd paragraph, 2nd sentence, the information authors mention was never presented in text. There is no figure nor table showing statistical comparison between CPE exposed animals and controls, so the authors should either present it first in the results (or with previously suggested supplementary tables) or remove this from discussion.
9. The same goes for the 3rd sentence in which authors state the differences between controls and clindamycin alone. Did authors refer here to comparison between CPE and CPE+clinda, since they do mention Tannerellaceae and Muribaculaceae?
10. In Discussion, 2nd paragraph, 4th sentence authors say: "A hallmark of CPE colonisation compared to clindamycin alone was the near disappearance of Muribaculaceae, whereas Tannerellaceae expanded further". I believe here the authors mean that Akkermansiaceae expanded and not Tannerellaceae.
11. The last 2 sentences of the 2nd paragraph of discussion should be re-phrased or removed. The relationship between Barnesiella and CPE in patients is correlational, the phylogenetic similarity between Barnesiella and Muribaculaceae is relative and thus this sentence sounds a bit speculative.
12. In Methods section- Lung injury, authors should indicate when the procedure was done- if it was 18h after infection.
13. In Methods section- Microbiome and SCFA analysis the sentence goes: "Phylogenetic data clustered into ASVs were further analysed using the phyloseq package...". I believe authors meant to say: "ASVs clustered in phylogenetic levels...".
14. Figure 1: if possible, include all available animals for microbiome analysis- in the plot 1b were 11-13 mice per group, while in 1d there are only 5. Also, it would be relevant to state in figure legend if mice whose microbiome was analyzed were all coming from the same experiment, or the authors included representatives from different batches.
15. Could the authors confirm that they applied FDR when performing statistical comparison between microbiomes of different experimental groups. This should be also stated in the legend.
16. I noticed in Figure 1d in section where comparison between clindamycin and CPE+clindamycin is plotted, that there is no difference in abundance of Enterobacteriaceae family. Is this an error? Could the authors discuss this?
17. Figure 2- Could the authors state in legend of the figure which statistical tests were applied here?
18. Figure 4- it would be helpful if the authors included mortality for CPE alone group here.
19. Figure 6- the color code for bar plot is different from the one used in Figure 1. Could the authors correct this so it is easier for the reader to compare results between figures?
20. Could the authors explain the difference between spleen counts in CPE colonized animals plotted in Figure 2b and Figure 7c. It is surprising that the hallmark of CPE colonization was higher rate of P.

aeruginosa dissemination into the spleen as compared to controls, which is presented in Figure 2b. But in figure 7c the CFUs in spleen are the same between control and CPE colonized animals.

21. Could the authors discuss the possible mechanisms of interaction between CPE and Muribaculaceae and Akkermansiaceae that would explain the altered abundances of these gut bacteria when mice are colonized with CPE?

Reviewer #2 (Remarks to the Author):

This is an original study by Le Guern and colleagues where they describe the effect of gut colonization with carbapenemase-producing *Klebsiella pneumoniae* on *Pseudomonas aeruginosa* lung infection. They conclude that gut colonization by CPE results in a specific gut dysbiosis and increases the severity of lung infection.

Although multiple studies have described the effects of (antibiotic-induced) gut microbiota disruptions and SCFA supplementation on pulmonary infections (e.g. Schuijt et al. *Gut* 2016, PMID 26511795; Sencio et al. *Cell Rep* 2020, PMID: 32130898; Trompette et al. *Immunity* 2018, PMID 2976810), this is - to the best of my knowledge - the first manuscript describing the effects of gut dysbiosis caused by colonization with multidrug-resistant bacteria on pneumonia.

In general, I find this an interesting manuscript and the authors' conclusions are supported by the data. Other strengths include the clear rationale and good experimental design. However, I have some comments as outlined below:

Comments:

1. From the figures, the group sizes are unclear to me. Figure 1 shows data of only 5 animals per group, while figures 2 and 3 includes data of 9-11 animals per group. Moreover, based on figure 6, it seems that each group comprised 18 animals. However, in figures 4 and 5 data of 14 or 15 animals is shown. Could the authors clarify these differences in group sizes and include them in both the methods and results section?

2. There seems a difference in the effect of CPE+Clinda between the experiments. Figure 1D shows that all CPE+Clinda-treated animals have an increased relative abundance of Enterobacteriaceae (as you would expect in CPE gut colonization). However, from Figure 6B the effect of CPE+Clinda treatment on the relative abundance of Enterobacteriaceae seems much smaller, or sometimes even absent. Could the authors elaborate on these differences?

3. In my opinion, an important conclusion of this manuscript seems that the effect of CPE colonization on lung infection was "due to the specific dysbiosis associated with CPE implementation rather than the presence of CPE" (lines 219-220). This finding could suggest that reconstitution or SCFA

supplementation would be a better future therapeutic intervention than decolonization of CPE. It would be preferable if this conclusion is included in the abstract. As written, it might give the impression that CPE colonization directly worsens lung infection.

4. It is noteworthy that antibiotic treatment did not result in worse outcomes of lung infection as it did result in gut dysbiosis (as shown in Figure 1) and the effects of antibiotic-induced gut microbiota disruptions have been extensively described. In my opinion, the authors should discuss this unexpected result and its possible causes.

5. Details for the measurement of bacterial load in the lung and spleen were not described in the methods section.

Reviewer #3 (Remarks to the Author):

In this manuscript, Guern et al tried to show that gut colonization with multidrug-resistant *Klebsiella pneumoniae* leads to worse lung infection with *Pseudomonas aeruginosa*. Although offering some interesting observations, the study at current form is rather preliminary without solid mechanistic investigation. The data do not solidly support the conclusion in general. Although the authors showed a decrease of Muribaculaceae in mice with antibiotics and CPE exposure, there was no data showing it is the decreased Muribaculaceae mediates the CPE exposure effect. The same thing for SCFA. Although the authors showed a lower SCFA level in mice with antibiotics and CPE exposure and supplementation with SCFA improved the lung infection, FMT did not increase SCFA level, thus it is hard to convince that decreased SCFA was responsible for the CPE exposure effect. Some major concerns need to be addressed to further improve the quality of the manuscript:

- 1) Fig 2b only showed an increase of CFU in the spleen but not in the lung. The lung is actually the site of infection. Not clear how such changes in the spleen would affect lung infection.
- 2) Fig 3a and Supple Fig 1, please show FACS profiles for all those cell types.
- 3) Fig 3b and Fig 5b, was the TNF level increased in the CPE+Clinda group compared to the control group?
- 4) Fig 5b, FMT actually decreased IL-22. Not clear what this means as IL-22 in general functions as protective for lung infection.

Response: First, we would like to thank the reviewers for their careful reading of our manuscript. We appreciate and have taken into account all the comments and suggestions. We believe this has improved the quality and clarity of the manuscript and that the additional experiments strengthen the conclusions of our manuscript.

REVIEWER COMMENTS

Reviewer #1 (Remarks to the Author):

In the manuscript titled “Gut colonisation with multidrug-resistant *Klebsiella pneumoniae* worsens *Pseudomonas aeruginosa* lung infection” authors seek to determine the effect of asymptomatic gut colonization with multidrug-resistant *Klebsiella pneumoniae* (CPE) on outcome of lung infection provoked by *P. aeruginosa*. The authors first established a model of stable gut colonization with CPE by treating mice with clindamycin and exposing them to CPE in drinking water. After this, they infect mice with *P. aeruginosa* and follow the infection progression. The authors characterized microbiome changes, disease and histological scores, and lung immune cells profile as a result of *P. aeruginosa* infection in CPE colonized mice. They saw that CPE colonized and *P. aeruginosa* infected mice had higher disease score than animals that weren't CPE colonized. They also identified members of the gut microbiome whose abundances were altered in CPE colonized and *P. aeruginosa* infected mice which lead them to hypothesize that these bacteria (specifically Muribaculaceae) might be responsible for worse outcome of lung infection in CPE colonized animals. To test this, authors performed faecal microbiome transplant in CPE colonized and *P. aeruginosa* infected mice which resulted in lower disease score and mortality as compared to mice that were colonized and infected but didn't receive FMT. FMT also affected lung immune profile. The authors further hypothesized that altered abundance of Muribaculaceae in the gut of CPE colonized mice was resulting in lower short-chain fatty acid (SCFA) abundance and was the reason for more severe lung infection in these animals. They tested the role of SCFA administration on mortality and disease score in CPE colonized and *P. aeruginosa* infected animals and saw reduction in both.

The role of gut microbiome dysbiosis on host's health is a subject of great interest and different aspects of it have been widely studied. The authors take on this, by studying the effect of CPE gut colonization on outcome of *P. aeruginosa* provoked lung infection, is novel. However, there are several major points that would need to be addressed for the manuscript to be published.

Major points:

1. It would be important to know if the effect observed by CPE colonization of the gut on lung infection outcome is specific for the strain authors used or it could be reproduced with other members of Enterobacteriaceae family.

Response: We thank the reviewer for sharing this important concern. We performed additional experiments with two other strains of *Klebsiella pneumoniae* producing other types of carbapenemases: KPC (strain KPL0.3, Genbank JANLCP000000000.1) and OXA-48 (strain KPL0.4, Genbank JANLDI000000000.1). These strains were whole genome sequenced for this project. Both strains lead to stable gut colonization after clindamycin injection, and lung infection was also more severe following gut

colonization (Supplementary Figure 1). Thus, the worse lung infection outcomes were not specific to the strain of carbapenemase-producing *K. pneumoniae* (NDM) previously used, as assessed with other strains of *K. pneumoniae* producing different carbapenemases.

We focused on *K. pneumoniae* as indicated in the title of the study, and we added in the discussion that the results may be different for other members of the *Enterobacteriales*. Additional studies requiring specific models are required to evaluate the impact of colonization with other members of the *Enterobacteriaceae* family

Changes: Supplementary Figure 1 was added, where we evaluated the effect of two other strains of *K. pneumoniae* producing KPC or OXA-48 carbapenemases.

Results: “To ensure that the worse lung infections outcomes were not specific to the CPE strain used (*K. pneumoniae* producing NDM carbapenemase), we reproduced these results with two other strains (*K. pneumoniae* producing OXA-48 or KPC carbapenemases) (Supplementary Figure 1). Stable gut colonisation by CPE was maintained up to day 14. Clinical disease severity score, alveolar-capillary permeability (lung injury index), and dissemination of *P. aeruginosa* into the spleen were increased for mice colonised by *K. pneumoniae* producing OXA-48 or KPC carbapenemases” (Lines 112-118)

Discussion: “This study has several limitations. First, we focused only on *K. pneumoniae* and did not evaluate the effects of colonisation with other members of the *Enterobacteriaceae* family. (Lines 319-320)

2. The authors were focusing on Muribaculaceae as potentially relevant microbiome members whose depletion led to worse lung infection outcomes. However, these bacteria are not the only ones with altered abundances in CPE colonized mice. The role of Akkermansiaceae and Sutterellaceae was not investigated nor discussed even though these were also altered in CPE colonized mice. In fact, Akkermansiaceae were increased in CPE colonized mice, but upon FMT their levels decreased, suggesting these can also explain the lung immune phenotype authors observed. Since members of this family are known for colonizing the mucus of the gut and impacting the immune response of the host it is of high importance to investigate their role in this model.

Response: We agree with the reviewer that other bacteria than *Muribaculaceae* were also modified for mice colonized by CPE. We updated the microbiome analysis as suggested in minor comment number 1 and provided comprehensive tables of the taxa significantly modified between groups using DESeq2 (Supplementary Tables 1 to 4). *Akkermansiaceae* were significantly increased in mice colonized by CPE (CPE+Clinda group) compared to control group (Supplementary Table 1), but there were no significant differences between mice colonized by CPE (CPE+Clinda group) and mice exposed to clindamycin (Clinda group) ($p=0.56$) or after FMT ($p=0.44$). *Muribaculaceae* abundance, however, was significantly altered for all the previous comparisons. The abundance of other strictly anaerobic bacteria was also altered, depending on the group (Supplementary Tables 1 to 4).

We apologize for not providing detailed comprehensive tables of the microbiome analysis in the previous version of the manuscript. We nonetheless added a paragraph

to discuss the increased abundance of *Akkermansiaceae* in mice exposed to clindamycin or colonized by CPE, given the important role of *Akkermansiaceae* in immunomodulation described in the literature. We believe that increase in *Akkermansia* relative abundance may be explained by antibiotic use as previously reported, but further studies are warranted to evaluate the impact of *Akkermansia* overabundance on the worse lung outcomes.

Changes: Figures 1 and 6 were updated, Supplementary Tables 1 to 4 were added.

Results: "Gut colonisation by CPE was related to a specific dysbiosis characterised by a consistently marked decrease in *Muribaculaceae*, *Rikenellaceae*, and *Lachnospiraceae_NK4A136_group* compared to gut dysbiosis associated with clindamycin alone." (Lines 81-84)

Discussion: "*Akkermansiaceae* relative abundance increased in mice exposed to clindamycin (median 5.8%) or colonised by CPE (median 9.9%) compared with controls (median 0.1%) and tended to decrease after FMT (median 0.6%). *Akkermansia muciniphila* colonises the mucus of the gut and is considered to present beneficial anti-inflammatory effects. However, an overabundance of *Akkermansia* (defined as > 4.8%) was associated with worse overall survival in patients with advanced cancer. Moreover, antibiotic use was associated with a relative dominance of *Akkermansia* in this cohort. In our model, an increase in *Akkermansia* may be explained by antibiotic use, but further studies are warranted to evaluate the impact of *Akkermansia* overabundance on the worse lung outcomes." (Lines 243-252)

3. The evidence pointing to SCFAs as responsible for lung phenotype and infection outcome is inconclusive: the administration of SCFAs to CPE colonized animals does have an ameliorating effect on infection outcome, however this seems to be unrelated to previously described results pertaining to this animal model. The animals that received clindamycin only (without CPE exposure) had similar levels of SCFA to CPE colonized animals although their outcome parameters were pronouncedly different, suggesting SCFA alone are not responsible for the higher mortality and disease score in CPE colonized mice. Moreover, it would be necessary to see lung immune profile in SCFA treated animals and compare it to CPE colonized and FMT treated animals to be able to make conclusions on whether SCFA are in fact restoring lung alterations caused by CPE colonization. Also, readers need to be provided with SCFA levels upon their administration so that those can be compared to SCFA levels in CPE colonized and FMT treated animals.

Response: We thank the reviewer for pointing that decreased levels of SCFA alone may not be responsible for the lung phenotype in our model, even if SCFA can participate in this phenomenon.

We performed additional experiments to increase the number of samples for the comparison of SCFA levels (updated Figure 7). The difference between CPE colonized mice (CPE+Clinda) and clindamycin only mice (Clinda) is now significant for total levels of SCFA (20.6 vs 39.0 $\mu\text{mol/g}$, $p = 0.05$) and acetate (16.4 vs 29.5 $\mu\text{mol/g}$, $p = 0.001$).

We analyzed SCFA levels in mice treated by SCFA supplementation. SCFA levels were higher for SCFA supplemented mice compared to CPE+Clinda mice for total SCFA levels (36.6 vs 22.6 $\mu\text{mol/g}$, $p = 0.02$) and acetate (29.1 vs 18.6 $\mu\text{mol/g}$, $p = 0.01$), but there was no difference for propionate and butyrate levels (Supplementary Figure 6). We analyzed the SCFA levels in the caecum lumen, and it was previously reported that butyrate is quickly absorbed before reaching the colon (PMID: 17373743). Although we did not obtain similar levels of SCFA as in control mice for propionate and butyrate, SCFA supplementation significantly reduced disease score and lung injury, as was already shown in our study with FMT.

We performed an additional experiment to analyze the immune profile of SCFA treated mice. Our data showed that SCFA had a limited effect on the recruitment of immune cells in the lungs and the spleen (Supplementary Figure 8 in lung and 9 in spleen). Previous reports demonstrated that SCFA had increased bacteria endocytosis and cell activation (PMID: 32130898 and 30683619). Altogether, our data suggest that the SCFA supplementation restored an efficient antibacterial response which may have been through activating resident cells and limiting the alteration of the alveolocapillary barrier (as assessed by reduction of the alveolar-capillary permeability) rather than through recruitment of immune cells. It was previously reported that propionate and butyrate restored epithelium barrier function in the airways (PMID: 33374733).

Changes: Figure 7 was updated. Supplementary Figures 8 and 9 were added.

Results: “Acetate, propionate, butyrate and total SCFA concentrations in caecum were decreased in mice colonised by CPE compared to those only exposed to clindamycin (total SCFA concentration $20.6 \pm 3.3 \mu\text{mol/g}$ versus $39.0 \pm 16.3 \mu\text{mol/g}$, respectively, $p=0.049$) (Figure 7a). There was no significant difference in the total SCFA concentration after FMT, although there was a trend for increased levels (total SCFA concentration $20.6 \pm 3.3 \mu\text{mol/g}$ for mice colonised by CPE versus $38.3 \pm 12.0 \mu\text{mol/g}$ following FMT, $p=0.063$). Nevertheless, FMT significantly increased the levels of acetate and propionate (Figure 7a).” (Lines 194-201)

“SCFA supplementation significantly increased the concentrations in the caecal lumen for total SCFA and acetate, but there was no significant difference for propionate and butyrate levels (Supplementary Figure 7).” (Lines 204-207)

“We analysed the lung immune profile of SCFA-treated mice by flow cytometry. SCFA had a limited effect on the recruitment of immune cells in the lungs and the spleen (Supplementary Figures 8 and 9).” (Lines 214-216)

Discussion: “Moreover, propionate and butyrate increased the barrier of airway epithelial cells. In our model, SCFA restored epithelial barrier function (as assessed by alveolar-capillary permeability) without modification of the bacterial load. SCFA had a limited effect on the recruitment of immune cells in the lungs and the spleen, in contrast with FMT. Our data suggest that SCFA supplementation restored an efficient antibacterial response, possibly by activating the resident cells, and limited the alteration of the alveolar-capillary barrier rather than by the recruitment of immune cells.” (Lines 309-316)

4. The relationship between Muribaculaceae abundance and SCFAs wasn't proved. The authors suggested that decrease in Muribaculaceae in CPE colonized animals was the reason for SCFA depletion. It would be relevant to perform experiments in

which CPE colonized animals would be treated with a member of Muribaculaceae family and disease outcome and SCFA levels would be followed. In fact, this should be performed also for a member of Akkermansiaceae as well. Confirming the role of these gut bacteria on lung infection outcome before diving into the mechanism of action is very important for making solid conclusions.

Response: We thank the reviewer for pointing out that the role of individual gut bacteria was not proven in our model.

We performed additional experiments by supplementing mice colonized by CPE (CPE+Clinda) with the reference strain of *Muribaculum intestinale* DSM 28989 (belonging to the family *Muribaculaceae*). These mice received daily orogastric gavage with a suspension of *M. intestinale* for the 3 days preceding infection. Our data showed that this treatment did not significantly modify the worse lung infection outcomes associated with CPE colonisation (Supplementary Figure 5a-c). However, we could not obtain gut colonization with *M. intestinale* despite iterative orogastric gavages (Supplementary Figure 5d), in contrast with the data reported after FMT (Figure 6).

Recently, Chng et al. evaluated the gut microbiota recovery following antibiotic therapy in a murine model (PMID: 32632261). They showed that most of the species producing SCFA were considered as secondary or tertiary species that need breakdown products from primary species. It is possible that *Muribaculaceae* are not able to colonize the gut when they are given individually, but that colonization is possible following FMT because a diverse ecosystem is provided. We added this point in the discussion.

Changes: Supplementary Figure 5 was added.

Results: “Supplementation in *Muribaculum intestinale* alone does not lead to stable colonization

Given the reduced abundance in *Muribaculaceae* for mice colonised by CPE, we evaluated the effects of supplementation in *Muribaculum intestinale* DSM 28989 by daily orogastric gavage for the three days preceding infection. Supplementation with *M. intestinale* did not significantly modify the worse lung infection outcomes associated with CPE colonisation (Supplementary Figure 5). *M. intestinale* supplementation did not result in stable gut colonisation by *Muribaculum*, as the relative abundance in *Muribaculum* sp. assessed by qPCR on Day 14 represented 7.9% for the control group, 0.009% for mice colonized by CPE, and 0.006% for mice supplemented in *M. intestinale* (Supplementary Figure 5), in contrast with the data reported after FMT (Figure 6).” (Lines 179-189)

Discussion:

“We tried to evaluate the effect of specific supplementation in *Muribaculum intestinale* by iterative orogastric gavage but did not obtain stable gut colonisation with *M. intestinale*. Recently, Chng et al. evaluated the gut microbiota recovery following antibiotic therapy in a murine model. They showed that most of the bacterial species producing SCFA were considered secondary or tertiary species that need breakdown products from primary species. In our model, colonisation was possible following FMT because a diverse ecosystem was provided, confirming that *Muribaculaceae* are not able to colonise the gut when they are given individually.” (Lines 287-294)

Minor points:

1. Microbiome analysis could be more extensive. It would be nice if authors could provide comprehensive tables of all taxa passing FDR when different experimental groups are compared. This could be done for both figures 1 and 6 and provided as supplementary tables. Also, it would be easier for the reader to understand differences between microbiome of different experimental groups represented in Figure 1 if the authors provided PCoA biplot instead of 2 separate graphs with differences between CPE and CPE+clindamycin; and clindamycin and CPE+clindamycin. This biplot should include also control group of animals.

Response: We improved microbiome analysis. DESeq2 was used to perform differential abundance analysis, correcting the p-value for multiple testing using the Benjamini-Hochberg method, in Supplementary Table 1 to 4 (for Figures 1 and 6 and Genus and Family level). PCoA biplot and alluvial plots were also provided for Figures 1 and 6.

Changes: Figures 1 and 6 were updated, Supplementary Tables 1 to 4 were added.

2. Could authors please provide p values in text when describing results of statistical analysis?

Response: We added *P* values in text.

3. Authors should introduce all the figures in text. In many instances results are described without explaining where the result is plotted.

Response: We added an introduction of all the Figures in text.

4. Authors should introduce properly all the abbreviations. In several instances abbreviation was never introduced, or it was introduced later in text.

Response: We added introduction of all the abbreviations in text.

5. In the results section authors mention alveolar capillary permeability, while in the figures they plot % of lung injury. Could the authors clarify in text that this is the same thing.

Response: Indeed, alveolar capillary permeability and percentage of the lung injury is the same notion, this has been clarified in the revised text.

Changes: “Outcomes of lung infection by *P. aeruginosa* were worsened in mice colonised by CPE as assessed by clinical disease severity score, alveolar-capillary permeability (lung injury index) (Figure 2a)” (Line 1000)

6. Results represented in Figure 3b were never mentioned nor explained in the text.

Response: We presented results from Figure 3b.

Changes: “The concentrations of TNF- α in the BAL were significantly increased in mice colonised by CPE compared to mice having received clindamycin (958.2 ± 265.5 versus 514.2 ± 512.4 pg/ml, $p=0.048$) (Figure 3b).” (Lines 136-138)

7. Authors suggested in text that CPE decontamination by FMT was unsuccessful, since CPE could be detected in feces a day after FMT. I suggest the authors to always refer to this time window, since it is unclear if the CPE levels would drop in FMT treated group in case the animals were followed beyond two days post FMT.

Response: We added that this was true only for this specific time window. Indeed, we assessed CPE levels only one day after the last FMT (or three days after the first FMT).

Changes:

Results: “Mean CPE load at day 14 (three days after the first FMT) was 8.0 ± 0.7 log CFU/g of stool for mice colonised by CPE and 8.1 ± 0.6 log CFU/g of stool following FMT ($p=0.54$).” (Lines 164-167)

“Overall, FMT partially restored the gut microbiota diversity and composition in mice colonised by CPE without reducing CPE loads (i.e., without ‘decolonising’ the gut) in this time window.” (Lines 176-178)

Discussion: “Of note, FMT did not act through ‘decolonisation’ of the gut because there was no CPE load decrease in stools within three days after the first administration.” (Lines 280-281)

8. In Discussion, 2nd paragraph, 2nd sentence, the information authors mention was never presented in text. There is no figure nor table showing statistical comparison between CPE exposed animals and controls, so the authors should either present it first in the results (or with previously suggested supplementary tables) or remove this from discussion.

Response: We added appropriate statistical comparison in Supplementary Table 1 (sheet “CPE vs Control”) and Supplementary Table 2 (sheet “CPE vs Control”), and an alluvial plot with mean relative abundance values per bacterial family in Figure 1 and Figure 6.

Changes: Supplementary Tables 1 to 4 were added, Figures 1 and 6 were updated.

9. The same goes for the 3rd sentence in which authors state the differences between controls and clindamycin alone. Did authors refer here to comparison between CPE and CPE+clinda, since they do mention Tannerellaceae and Muribaculaceae?

Response: We added appropriate statistical comparison in Supplementary Table 1 (sheet “Clinda vs Control”) and Supplementary Table 2 (sheet “Clinda vs Control”). We referred to the comparison between Clinda and Control. We modified the sentence to reflect the data from these Tables.

Changes: Supplementary Tables 1 to 4 were added.

“Alteration of the gut microbiota by clindamycin alone was associated with decreased α -diversity, expansion of *Tannerellaceae* and *Enterobacteriaceae* and decreased abundance in *Prevotellaceae*.” (Lines 231-233)

10. In Discussion, 2nd paragraph, 4th sentence authors say: “A hallmark of CPE colonisation compared to clindamycin alone was the near disappearance of Muribaculaceae, whereas Tannerellaceae expanded further”. I believe here the authors mean that Akkermansiaceae expanded and not Tannerellaceae.

Response: For CPE+Clinda versus Clinda, we show a decrease in *Muribaculaceae* and *Rikenellaceae* in CPE+Clinda. For *Lachnospiraceae*, some genera are decreased in CPE+Clinda (*Lachnospiraceae_NK4A136_group*) but others are increased (*Hungatella*). We modified the sentence according to Supplementary Tables 1-4. *Akkermansiaceae* tended to increase after clindamycin or CPE colonisation, but the difference between CPE+Clinda versus Clinda was not significant.

Changes: Supplementary Tables 1 to 4 were added.

“A hallmark of CPE colonisation compared to clindamycin alone was the near disappearance of *Muribaculaceae* and *Rikenellaceae*, associated with modifications in *Lachnospiraceae* (increase in *Hungatella* but decrease in *Lachnospiraceae_NK4A136_group*).” (Lines 236-239)

11. The last 2 sentences of the 2nd paragraph of discussion should be re-phrased or removed. The relationship between *Barnesiella* and CPE in patients is correlational, the phylogenetic similarity between *Barnesiella* and Muribaculaceae is relative and thus this sentence sounds a bit speculative.

Response: We agree with the reviewer that this is speculative and have removed the last two sentences of the paragraph.

Changes: The two sentences were removed. (Line 242)

12. In Methods section- Lung injury, authors should indicate when the procedure was done- if it was 18h after infection.

Response: Lung injury was indeed 18h after infection; we have modified the Methods.

Changes: “Alveolar-capillary membrane permeability was evaluated using fluorescein isothiocyanate (FITC)-labelled albumin leakage from the vascular to the alveolar-interstitial compartment 18 hours after infection.” (Lines 408-410)

13. In Methods section- Microbiome and SCFA analysis the sentence goes: “Phylogenetic data clustered into ASVs were further analysed using the phyloseq package...”. I believe authors meant to say: “ASVs clustered in phylogenetic levels...”.

Response: We have modified this sentence as suggested.

Changes: “ASVs clustered in phylogenetic levels were further analysed using the phyloseq package” (Lines 439-440)

14. Figure 1: if possible, include all available animals for microbiome analysis- in the plot 1b were 11-13 mice per group, while in 1d there are only 5. Also, it would be relevant to state in figure legend if mice whose microbiome was analyzed were all coming from the same experiment, or the authors included representatives from different batches.

Response: We already added all the animals available for analysis. For microbiome analysis, we have 5 animals per group from the same batch in Figure 1, and 18 animals per group from 4 different batches in Figure 6. We modified the legend from Figure 1 and 6.

Changes: Figure 1 “Gut microbiota 16S rRNA analysis at day 14 (5 mice per group from the same experiment)” (Line 632)
Figure 6 “Gut microbiota 16S rRNA analysis (18 mice per group representatives from 4 experiments)” (Line 669)

15. Could the authors confirm that they applied FDR when performing statistical comparison between microbiomes of different experimental groups. This should be also stated in the legend.

Response: We applied false discovery rate in Supplementary Tables 1 to 4 (Benjamini-Hochberg method). We added this in the Methods.

Changes: Supplementary Tables 1 to 4 were added.
Methods: “Differential abundance testing was performed using DESeq2 and applying Benjamini-Hochberg False Discovery Rate.” (Lines 443-445)

16. I noticed in Figure 1d in section where comparison between clindamycin and CPE+clindamycin is plotted, that there is no difference in abundance of Enterobacteriaceae family. Is this an error? Could the authors discuss this?

Response: We thank the reviewer for this remark. *Enterobacteriaceae* are increased in mice having received clindamycin compared with controls because the levels of *Escherichia coli* are increased. Control mice are often colonized with low levels of *E. coli* and *E. coli* is naturally resistant to clindamycin. At the genus level, *Escherichia/Shigella* were increased in mice exposed to clindamycin compared to controls, whereas both *Klebsiella* and *Escherichia/Shigella* were increased in CPE colonised mice compared to controls (Supplementary Table 2). However, using microbiome analysis, identification of *Enterobacteriaceae* to the genus level is difficult just by sequencing only the V6-V8 region of the 16S rRNA gene (300 bp paired-end) because of considerable similarity.

We confirmed this in an additional experiment with 6 mice per group by performing microbiological culture. Average bacterial load in *E. coli* at Day 14 was 3.7 log CFU / g of stool sample for control mice, 8.7 log for clindamycin mice, and 8.2 log for CPE+Clindamycin. For *K. pneumoniae*, we found 8.9 log for CPE+Clindamycin group and 0 for the other groups.

Changes:

Results: "Relative abundance in *Enterobacteriaceae* was increased in mice exposed to clindamycin or colonised by CPE compared with controls (Supplementary Table 1). At the genus level, *Escherichia/Shigella* were increased in mice exposed to clindamycin compared to controls, whereas both *Klebsiella* and *Escherichia/Shigella* were increased in CPE colonised mice compared to controls (Supplementary Table 2). By conventional microbiological culture, *Escherichia coli* levels were increased in mice exposed to clindamycin (8.7 log colony-forming units (CFU)/g of stool) or colonised by CPE (8.2 log CFU/g of stool) compared with controls (3.7 log CFU/g of stool), whereas *K. pneumoniae* was only recovered in mice colonised by CPE (8.9 log CFU/g of stool)." (Lines 84-93)

Discussion: "Higher *Enterobacteriaceae* abundance in the clindamycin group compared to control was explained by an increase in the levels of commensal *Escherichia coli* as assessed by conventional microbiological culture." (Lines 233-236)

17. Figure 2- Could the authors state in legend of the figure which statistical tests were applied here?

Response: Continuous variables were compared using ANOVA followed by a Tukey post hoc test as stated in the Methods. Survival curves were analysed using the log-rank (Mantel-Cox) test. We described statistical tests in the Methods section.

18. Figure 4- it would be helpful if the authors included mortality for CPE alone group here.

Response: We performed additional experiments to add the mortality for 15 mice from the CPE alone group. *P* value for this figure was updated to 0.03 instead of 0.02 (Log-rank test).

Changes: Figure 4d was updated to add the CPE alone group in the survival analysis.

19. Figure 6- the color code for bar plot is different from the one used in Figure 1. Could the authors correct this so it is easier for the reader to compare results between figures?

Response: We have modified the colors from the alluvial plot to maintain consistency between Figure 1 and Figure 6.

Changes: Figure 1 and Figure 6 were updated.

20. Could the authors explain the difference between spleen counts in CPE colonized animals plotted in Figure 2b and Figure 7c. It is surprising that the hallmark of CPE colonization was higher rate of *P. aeruginosa* dissemination into the spleen as compared to controls, which is presented in Figure 2b. But in figure 7c the CFUs in spleen are the same between control and CPE colonized animals.

Response: In Figure 7c, spleen dissemination is not higher for mice colonized by CPE, contrary to what was shown in Figure 2b and Figure 4b. This may be due to the inherent variability of an *in vivo* model.

We performed additional experiments with two other *K. pneumoniae* strains (KPC and OXA48) and we showed that CPE colonization was also associated with an increased dissemination of *P. aeruginosa* in the spleen (Supplementary Figure 1), confirming the data reported in Figure 2b and Figure 4b. In addition, higher dissemination of *P. aeruginosa* in the spleen is also consistent with increased lung injury index (reflect of the alveolar-capillary permeability). Lung injury index is consistently increased in Figure 2a, Figure 4a, Figure 7b and Supplementary Figure 1.

21. Could the authors discuss the possible mechanisms of interaction between CPE and Muribaculaceae and Akkermansiaceae that would explain the altered abundances of these gut bacteria when mice are colonized with CPE?

Response:

Muribaculaceae are able to produce SCFA from fibers and they are present in high abundance in the gut of control mice. In vitro, SCFA directly inhibited the growth of antibiotic-resistant *Enterobacteriales*, including carbapenemase-producing *K. pneumoniae* (Sorbara et al., PMID: 30563917). There may be a competition between CPE and bacteria producing SCFA.

We also added a paragraph on increased relative abundance of *Akkermansia* in Clinda and CPE+Clinda groups, which is probably linked to antibiotic use as reported previously.

Changes:

Discussion: "In vitro, SCFA also directly inhibited the growth of antibiotic-resistant *Enterobacteriales*, including carbapenemase-producing *K. pneumoniae*." (Lines 297-299)

Reviewer #2 (Remarks to the Author):

This is an original study by Le Guern and colleagues where they describe the effect of gut colonization with carbapenemase-producing *Klebsiella pneumoniae* on *Pseudomonas aeruginosa* lung infection. They conclude that gut colonization by CPE results in a specific gut dysbiosis and increases the severity of lung infection. Although multiple studies have described the effects of (antibiotic-induced) gut microbiota disruptions and SCFA supplementation on pulmonary infections (e.g. Schuijt et al. Gut 2016, PMID 26511795; Sencio et al. Cell Rep 2020, PMID: 32130898; Trompette et al. Immunity 2018, PMID 2976810), this is - to the best of my knowledge - the first manuscript describing the effects of gut dysbiosis caused by colonization with multidrug-resistant bacteria on pneumonia. In general, I find this an interesting manuscript and the authors' conclusions are supported by the data. Other strengths include the clear rationale and good experimental design. However, I have some comments as outlined below:

Comments:

1. From the figures, the group sizes are unclear to me. Figure 1 shows data of only 5 animals per group, while figures 2 and 3 includes data of 9-11 animals per group. Moreover, based on figure 6, it seems that each group comprised 18 animals. However, in figures 4 and 5 data of 14 or 15 animals is shown. Could the authors clarify these differences in group sizes and include them in both the methods and results section?

Response: We stated the number of animals per group in the legend of each Figure. We didn't collect stools for microbiota analysis for every mouse due to the costs associated with Illumina Sequencing. For Figure 1, we analyzed stools from 20 mice (5 / group) from one experiment (littermates). For Figure 6, we analyzed stools from 72 mice (18 / group) representative from 4 different experiments. We added this precision to legend of Figure 1 and 6.

Changes:

We updated the legends of Figure 1 and 6.

Figure 1 "Gut microbiota 16S rRNA analysis at day 14 (5 mice per group from the same experiment)" (Line 631)

Figure 6 "Gut microbiota 16S rRNA analysis (18 mice per group representatives from 4 experiments)" (Line 669)

2. There seems a difference in the effect of CPE+Clinda between the experiments. Figure 1D shows that all CPE+Clinda-treated animals have an increased relative abundance of Enterobacteriaceae (as you would expect in CPE gut colonization). However, from Figure 6B the effect of CPE+Clinda treatment on the relative abundance of Enterobacteriaceae seems much smaller, or sometimes even absent. Could the authors elaborate on these differences?

Response:

We thank the reviewer for this remark. *Enterobacteriaceae* are increased in mice having received clindamycin compared with controls because the levels of *Escherichia coli* are increased. Control mice are often colonized with low levels of *E. coli* and *E. coli* is naturally resistant to clindamycin. At the genus level, *Escherichia/Shigella* were increased in mice exposed to clindamycin compared to controls, whereas both *Klebsiella* and *Escherichia/Shigella* were increased in CPE colonised mice compared to controls (Supplementary Table 2). However, using microbiome analysis, identification of *Enterobacteriaceae* to the genus level is difficult just by sequencing only the V6-V8 region of the 16S rRNA gene (300 bp paired-end) because of considerable similarity. We confirmed this in an additional experiment with 6 mice per group by performing microbiological culture. Average bacterial load in *E. coli* at Day 14 was 3.7 log CFU / g of stool sample for control mice, 8.7 log for clindamycin mice, and 8.2 log for CPE+Clindamycin. For *K. pneumoniae*, we found 8.9 log for CPE+Clindamycin group and 0 for the other groups.

We also updated Figure 1 and Figure 6 to include alluvial plots showing the mean of the relative abundance for each group. It is now easier to see that the relative abundance of *Enterobacteriaceae* is increased in Clinda, CPE+Clinda, or FMT groups compared with Control.

Changes:

Figures 1 and 6 were updated.

Results: "Relative abundance in *Enterobacteriaceae* was increased in mice exposed to clindamycin or colonised by CPE compared with controls (Supplementary Table 1). At the genus level, *Escherichia/Shigella* were increased in mice exposed to clindamycin compared to controls, whereas both *Klebsiella* and *Escherichia/Shigella*

were increased in CPE colonised mice compared to controls (Supplementary Table 2). By conventional microbiological culture, *Escherichia coli* levels were increased in mice exposed to clindamycin (8.7 log colony-forming units (CFU)/g of stool) or colonised by CPE (8.2 log CFU/g of stool) compared with controls (3.7 log CFU/g of stool), whereas *K. pneumoniae* was only recovered in mice colonised by CPE (8.9 log CFU/g of stool)." (Lines 84-93)

Discussion: "Higher *Enterobacteriaceae* abundance in the clindamycin group compared to control was explained by an increase in the levels of commensal *Escherichia coli* as assessed by conventional microbiological culture." (Lines 233-236)

3. In my opinion, an important conclusion of this manuscript seems that the effect of CPE colonization on lung infection was "due to the specific dysbiosis associated with CPE implementation rather than the presence of CPE" (lines 219-220). This finding could suggest that reconstitution or SCFA supplementation would be a better future therapeutic intervention than decolonization of CPE. It would be preferable if this conclusion is included in the abstract. As written, it might give the impression that CPE colonization directly worsens lung infection.

Response: We thank the reviewer for his suggestion and have modified the abstract accordingly.

Changes: "Asymptomatic gut colonisation by CPE led to a specific gut dysbiosis, increasing the severity of *P. aeruginosa* lung infection, with lower recruitment of alveolar macrophages and conventional dendritic cells. CPE associated dysbiosis was characterised by a near disappearance of the *Muribaculaceae* family and lower levels of short-chain fatty acids." (Lines 19-23)

4. It is noteworthy that antibiotic treatment did not result in worse outcomes of lung infection as it did result in gut dysbiosis (as shown in Figure 1) and the effects of antibiotic-induced gut microbiota disruptions have been extensively described. In my opinion, the authors should discuss this unexpected result and its possible causes.

Response: For antibiotic-induced gut dysbiosis, we used only two injections of one antibiotic, and waited one week after the last antibiotic administration before lung infection. Other published models such as Schuijt et al. (PMID: 26511795) use as much as four different antibiotics for a period of three weeks and wait only 2 days before lung infection. We have also published a similar effect of a treatment with Vanco and colistin and we demonstrated an effect of the dysbiosis one day after antibiotic administration. Interestingly, we have observed in this model, an effect of this dysbiosis on lung immune cells in uninfected mice whereas we have not detected this alteration in the present model (data not shown) confirming that the colonization-related dysbiosis have a different impact on Gut / Lung axis. We can suspect that in our model gut microbiota resilience is occurring in the Clinda or FMT group but is delayed in CPE+Clinda group.

Changes:

Discussion: "Indeed, other published models use as much as four different antibiotics for a period of three weeks and wait only two days before lung infection. Conversely, we showed that gut colonisation by CPE increased *P. aeruginosa* lung infection

severity. One hypothesis could be that gut microbiota resilience is occurring in the clindamycin group but is delayed in mice colonised by CPE.” (Lines 270-274)

5. Details for the measurement of bacterial load in the lung and spleen were not described in the methods section.

Response: We added details for measurement of bacterial load in the methods section.

Changes: “To evaluate the lung and spleen bacterial loads of *Pseudomonas aeruginosa*, the left lobe of the lung and spleen were collected and homogenised by bead-beating, then plated onto non-selective (LB agar without antibiotics) and selective medium (Cetrimide agar).” (Lines 389-392)

Reviewer #3 (Remarks to the Author):

In this manuscript, Guern et al tried to show that gut colonization with multidrug-resistant *Klebsiella pneumoniae* leads to worse lung infection with *Pseudomonas aeruginosa*. Although offering some interesting observations, the study at current form is rather preliminary without solid mechanistic investigation. The data do not solidly support the conclusion in general. Although the authors showed a decrease of Muribaculaceae in mice with antibiotics and CPE exposure, there was no data showing it is the decreased Muribaculaceae mediates the CPE exposure effect. The same thing for SCFA. Although the authors showed a lower SCFA level in mice with antibiotics and CPE exposure and supplementation with SCFA improved the lung infection, FMT did not increase SCFA level, thus it is hard to convince that decreased SCFA was responsible for the CPE exposure effect.

Some major concerns need to be addressed to further improve the quality of the manuscript:

1) Fig 2b only showed an increase of CFU in the spleen but not in the lung. The lung is actually the site of infection. Not clear how such changes in the spleen would affect lung infection.

Response: Both lung injury index (alveolar-capillary permeability) and bacterial dissemination (CFU in spleen) are increased in mice colonized by CPE+Clinda. These data suggest that the alteration of the alveolar-capillary barrier should facilitate the dissemination of the bacteria in the blood and the spleen. Confirming this, we have also detected more hemorrhage areas in the lung of CPE+Clinda mice by histological analysis. Moreover, we can also hypothesize that bacterial colonization affect the antibacterial defense in the spleen and so facilitating its development in this organ.

2) Fig 3a and Supple Fig 1, please show FACS profiles for all those cell types.

Response: We thank the reviewer for his suggestion. We have now included the gating strategy as Supplementary Figure 10 for both the lung and the spleen.

Changes: Supplementary Figure 10 was added.

Methods: “Phenotypes are shown in Supplementary Table 6 and gating strategy was reported in Supplementary Figure 10.” (Lines 428-429)

3) Fig 3b and Fig 5b, was the TNF level increased in the CPE+Clinda group compared to the control group?

Response: It was not significantly increased.

Fig3b: For TNF, ANOVA was $P=0.03$, post-hoc test was $P=0.048$ for CPE+Clinda versus Clinda, but only $P=0.065$ for CPE+Clinda versus Control.

Fig5b: For TNF, ANOVA was $P=0.0007$, post-hoc test was $P=0.0005$ for CPE+Clinda versus FMT, $P=0.0486$ for Control versus FMT, but only $P=0.2360$ for CPE+Clinda versus Control.

Changes:

Results: “The concentrations of TNF- α in the BAL were significantly increased in mice colonised by CPE compared to mice having received clindamycin (958.2 ± 265.5 versus 514.2 ± 512.4 pg/ml, $p=0.048$) (Figure 3b).” (Lines 136-138)

“In addition, FMT significantly decreased the concentrations of TNF- α in the BAL compared to colonised mice (335.2 ± 257.6 versus 720.0 ± 172.1 pg/ml, $p<0.001$) and IL-22 in comparison to control mice (31.6 ± 29.3 versus 52.8 ± 32.3 pg/ml, $p=0.04$) (Figure 5b).” (Lines 158-161)

4) Fig 5b, FMT actually decreased IL-22. Not clear what this means as IL-22 in general functions as protective for lung infection.

Response: Indeed, IL-22 has been reported as protective cytokine against *P. aeruginosa* (Mear et al., PMID: 24166952). We have also analyzed IL-17 expression and we did not observe a modulation of this cytokine. Our interpretation is related to the potential effect of SCFA on the ability of effector cells to endocytose and to kill bacteria. The lower level of IL-22 in this context might be linked to a lower activation of immune cells.

REVIEWERS' COMMENTS

Reviewer #1 (Remarks to the Author):

I thank the authors for providing this updated version of the manuscript. I am especially grateful for increasing the sample number in order to confirm that there is significant difference in total SCFA levels between CPE colonized mice and clindamycin treated mice (Figure 7). Overall, I believe that the authors addressed satisfactorily my comments and that the extensive work that has been done in this version has greatly improved the manuscript.